# Terahertz photonic heterodyne spectral analysis with (sub-) kHz resolution and 6.5 THz frequency coverage

Benedikt Krause [1] ✉, Sebastian Müller[2,3], Thomas Puppe [2], Lars Liebermeister [4], Garrit Schwanke [4], Milan Deumer [4], Robert Kohlhaas[4], Rafal Wilk[2], Nico Vieweg[2] & Sascha Preu [1] ✉

Spectrum analyzers and spectrometers are essential for designing sources, analyzing material properties, layer structures and fingerprinting substances. We present an ultra-wideband, continuous-wave photonic receiver with kHz-level spectral resolution in the terahertz domain for both heterodyne and homodyne detection. Employed as a spectrum analyzer front end, it records the emitted spectrum of a source under test, assessing spectral purity, spectral shape and undesired frequency components. It outperforms state-of-the-art electronic systems in terms of frequency coverage and system cost with a competitive spectral resolution and noise floor on the few aW/Hz level at room temperature. It covers the important frequencies above 1.5 THz, yet commercially inaccessible, where sources like quantum cascade lasers operate. When combined with a comb-based photonic source, we demonstrate hetero- and homodyne spectroscopy over an unprecedented frequency range from below 100 GHz to 6.5 THz and a very low noise floor. Locking the photonic system to GPS enables tracing back the measured parameters to SI units, being of key importance for metrological applications. The presented setups offer the broadest continuous-wave frequency coverage to date, combined with a sharp spectral resolution, enabling diverse applications ranging from fast non-destructive testing, astronomic high-resolution spectroscopy, to frequency-modulated RADAR.

Between optics and electronics, there is the terahertz (THz) frequency range, commonly defined as the frequency range between 100 GHz and 10 THz. Despite a vast growth of research activities in the past two decades, this spectral domain is still much less explored compared to optics and electronics, yet with a desperate need for more powerful sources and systems with high dynamic range and well-defined spectral properties. Key applications are future high-speed wireless communication, with several bands already defined in 6 G in the lower THz region[1] and more to come in follow-up standards, non-destructive testing and quality control[2], as well as spectroscopy[3] and imaging[4], with particular focus on medically or environmentally relevant topics. The latter includes, amongst others, breath gas analysis[5] for non-invasive diagnosis of illnesses, detection of pollutant trace gases[6], or medical imaging, e.g., of skin cancer[7]. A severe obstacle for approaching these applications on a larger scale is the lack of powerful, spectrally pure sources. In order to design, characterize and optimize these, in general, spectrum analyzers are used all across the electromagnetic spectrum from high frequency electronics (electronic

[1]Department of Electrical Engineering and Information Technology, Technical University of Darmstadt, Darmstadt, Germany. [2]TOPTICA Photonics AG, Gräfelfing, Germany. [3]Institute of Microwaves and Photonics, Friedrich Alexander University, Erlangen, Germany. [4]Fraunhofer Institute for Telecommunications, Heinrich-Hertz-Institute, Berlin, Germany. ✉e-mail: benedikt.krause@tu-darmstadt.de; sascha.preu@tu-darmstadt.de

spectrum analyzer, ESA) all the way to visible wavelengths (optical spectrum analyzer, OSA), including the Terahertz domain, where this manuscript will demonstrate a photonic spectrum analyzer (PSA) with exceptional specifications. In order to mitigate insufficient source power, coherent systems are frequently used. They benefit from suppressed out of band noise and usually achieve a much higher signal to noise ratio than their direct detection counterparts. Both heterodyne and homodyne photonic setups will be demonstrated in the following.

Historically, the THz domain was either approached by frequency-upscaled electronic devices[8] or by down-conversion of optical signals[9]. Electronic and photonic domains for THz generation and detection were clearly segregated. Nonlinear media and photomixers bridge this gap by combining electronic with photonic approaches. Nonlinear media benefit from high THz powers and large bandwidths, yet lack in efficiency. Resonators increase the coupling efficiency at the expense of tunability[10]. Photomixers, i.e., photodiodes (PDs) or photoconductors (see section photonic mixer in methods), use two laser signals for THz generation and detection at their difference frequency[11,12]. Meanwhile, photonic approaches transition into the electronic domains, e.g. in the form of a low phase noise photonic clock signal[13], or photonic local oscillators[14], and mixed electronic-photonic integrated circuits (EPICs)[15]. The highly important field of wireless THz communication benefits from opto-electronic devices directly transforming optical signals, transmitted through a fiber, to a wireless THz signal. However, engineering tools for performance analysis of THz devices, like vector network analyzers and spectrum analyzers, are still purely electronic. They consist of a microwave backbone system equipped with frequency extenders, currently available up to 1.5 THz. The extenders double or triple the microwave baseband frequency several times up to the desired THz frequency or from the THz frequency down to the base band[16]. These frequency multiplier chains are housed in metallic hollow core waveguides with limited bandwidths of about 50% of their center frequencies[17]. If a larger frequency range shall be analyzed, several extender bands have to be used. Covering the range of 100 GHz to 1.5 THz requires at least seven different extender chains, yet with strongly increasing cost with increasing frequency. Using several extenders also complicates the experiment as the setup must be altered and recalibrated for every extender band separately. In particular, revealing undesired harmonics is cumbersome. Nonetheless, the electronic system benefits from an Hz- to kHz-level spectral resolution and a very low noise floor.

Meanwhile, first alternatives to the electronic spectrum analyzer (ESA) have been demonstrated. For example, a Hilbert-transform spectrum analyzer based on a high-$T_c$ Josephson junction[18] achieves a high scanning speed of up to 4 THz/s and a frequency coverage of at least 1 THz, yet only a resolution of 50 MHz. It requires a cryogenic cooler to operate, limiting its operation to laboratory environments. Interferometer-based spectrum analyzers[19] cover a frequency range of 1.5 THz but only achieve a resolution of 3.6 GHz. Pulsed-comb-referenced spectrum analyzers[20,21] improve the resolution to the Hz level and cover typically 2-4 THz. Their limitation is the ambiguity of the measured signal requiring special pulsed systems with controllable repetition rates. Even then the system only manages to detect narrowband or single frequency signals. Frequency comb (FC) ptychoscopy[22] has the potential to cover a similar frequency range with a resolution in the kHz range while simultaneously enabling measurements of swiftly swept sources by implementing two frequency combs or a dual-frequency comb with different repetition rates and two detectors.

The ambiguity of pulsed comb-based systems can also be overcome by locking continuous-wave lasers to specific comb lines while tuning the comb's repetition rate, yet with a demonstrated resolution of 200 kHz and a tuning range up to 1.28 THz[23]. Likewise, high spectral purity can be achieved by locking the CW lasers to different resonances of a high quality factor cavity[24], with a sub-Hz frequency

stability demonstrated at 1.965 GHz and an accuracy of 6 kHz at 556.936 GHz. Coarse tunability of up to 1.2 THz is achieved by hopping between cavity resonances. The gaps between the resonances are closed by fine-tuning the difference frequency with a Mach-Zehnder modulator. Reference[25] references the CW lasers to a frequency comb, which itself is locked to an optical cavity achieving a terahertz frequency stability of 1.46 Hz. Removing the fiber thermal noise improves the stability further to 2 mHz. Due to the referenced frequency comb this setup is limited to THz frequencies spaced apart by the repetition rate of the FC, with a demonstrated frequency coverage of 1.1 THz. So far, these high-purity continuous-wave systems have not been demonstrated as spectrum analyzers and their maximum demonstrated operation frequency is, to date, below those of commercial ESAs (1.5 THz).

Recently, photonic heterodyne spectrum analyzers have been demonstrated[26–29]. These utilize continuous-wave (CW) lasers as a local oscillator (LO). An electro-optical comb based on an electro-optical phase modulator at telecom wavelengths covers a frequency range of up to 1 THz with a resolution 1 Hz[26]. Free-running distributed feedback (DFB) laser diodes enable a difference frequency coverage of up to 4.5 THz at a wavelength of 780 nm[29], a resolution of 1.2 MHz at a center wavelength around 1550 nm[27]. DFB laser diodes at 850 nm referenced to an electronic THz signal reduce the resolution to 100 Hz and the frequency coverage to 1 THz[28]. These systems rely on CW lasers as well as photoconductive mixers. The highest frequency coverage of these – used in a spectroscopy setup – to date is 5.5 THz[30,31].

Here we demonstrate a concept that combines the capability for heterodyne spectrum analysis and high-resolution spectroscopy up to 6.5 THz in a single measurement instrument while referencing the system to GPS for a high absolute frequency accuracy. The heterodyne photonic spectrum analyzer uses a newly established FC-referenced CW laser system and top-of-the-line photoconductive sources and receivers. The laser system has the capability of covering more than 10 THz[32] with competitive phase stability[33] at a Hz-level spectral resolution[34,35] at the lower end of the THz spectrum while sweeping with maximum speeds of 1 THz/s[32]. The combination with a photoconductor enables characterization of sources from the microwave range all the way up to 6.5 THz and possibly beyond. The speed of the system is eventually limited by post-detection electronics and evaluation, and the tuning speed of the phase-locked CW lasers. For high resolution spectroscopy, the system is expanded in two ways. First, with a photomixer hooked up to a photonic difference FC, phase-referenced to the photonic oscillator, and second with a PD hooked up to the FC-referenced CW laser system enabling a combined frequency coverage from below 100 GHz up to 6.5 THz, the largest frequency coverage of a CW photomixing system to date.

## Results
### Principle of operation
We employ the same photonic local oscillator (LO) laser signal and Rh:InGaAs[30] photoconductive receiver for both the heterodyne spectrum analysis as well as the high-resolution spectroscopy. The optical frequencies (~193 THz) of the lasers are significantly higher than the required LO frequency in the THz region. For the generation of the LO frequency, we employ a set of CW diode lasers (Toptica DL pro and CTL) referenced to an optical FC (Fig. 1a, Toptica DFC Core + ) generating two very pure optical tones with one of them tunable over a large range[32–35]. Heterodyning the two laser signals results in an intensity modulation at their frequency difference, i.e., the LO frequency. A photoconductor (Fig. 1b) absorbs the intensity-modulated light, causing the generation of electron-hole pairs at the same frequency and thus modulation of the conductance at the LO frequency. At the same time, an antenna attached to the photoconductor collects the THz signal of a source, generating a respective voltage at the photoconductor electrodes. The THz signal voltage is multiplied by

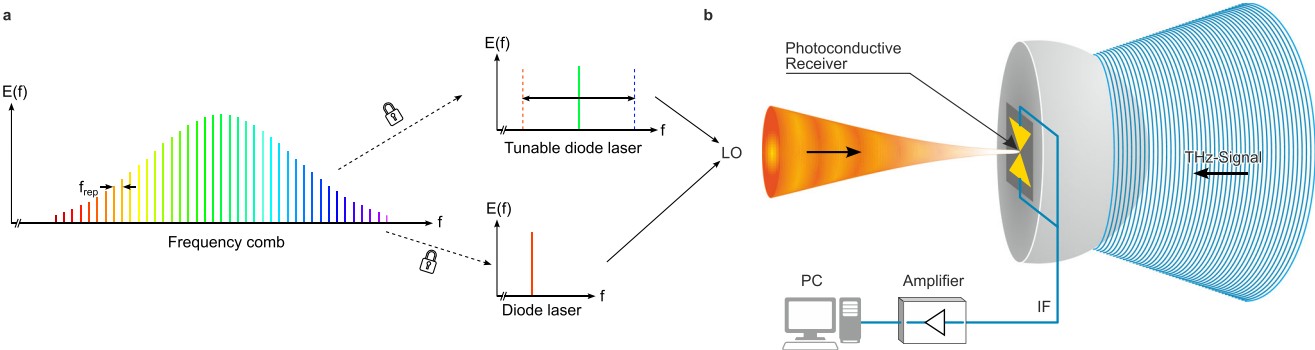

**Fig. 1 | Photoconductive terahertz downconversion. a** Spectral view of the two-tone generation for the local oscillator (LO) with two CW diode lasers – one tunable and one stationary – locked to the same frequency comb (FC). $f_{rep}$ denotes the repetition rate of the FC. **b** Schematic of the photoconductive receiver on a Silicon lens receiving the laser signals from the LO as well as the THz signal emitted by a source. The down-converted signal is transported at the intermediate frequency (IF), here in kHz to MHz range.

the varying conductance according to Ohm's law resulting in a difference frequency output current. If the LO is tuned close to the source frequency, the frequency of the current, termed intermediate frequency (IF) will be small enough in order to be detected with low-noise electronics. The photoconductor essentially mixes the envelope of the laser signal with the incoming THz-signal. In order to approach the thermal noise floor of the photoconductor, we employ a transimpedance amplifier (TEM Messtechnik PDA-S or Femto DHPCA-100) at gains between $10^6$ and $10^7$ V/A prior to digitizing the signal. Common spectrum analyzers show the power spectral density of this current, which is proportional to the power spectral density of the source. By simply sweeping the LO frequency, we can record the spectrum of any unknown source under test at a resolution defined by the spectral purity of the FC-referenced LO. Given that the source's power is high enough to be detected, the maximum tuning range is limited only by the tuning range of the LO (here >10 THz)[32] and the sensitivity of the photoconductor, which becomes smaller the higher the frequency. The frequency coverage of the used photoconductors is limited by the Reststrahlenband of the InP substrate to 7 THz. Removing the substrate allows for potential coverage within or possibly even beyond the Reststrahlenband of InP and InGaAs[36].

Within all measurements shown in this manuscript, the LO is additionally referenced via an oven-controlled crystal oscillator (OCXO) to a global positioning system (GPS) signal. The OCXO generates an 800 MHz reference signal that phase-locks to the 4th harmonic of the repetition rate of the FC. The additional referencing of the OCXO to the GPS gives the LO absolute frequency accuracy which we verify in a later section.

### Spectrum analyzer for unreferenced sources
Spectrum analyzers are mostly used to measure signals that originate from an external source that is not referenced to the analyzer itself. We showcase a PSA by characterizing the spectra emitted from two different THz-sources. The first source is a CW waveguide-integrated (WIN)-photodiode (PD)[31] from the Fraunhofer Heinrich-Hertz-Institute driven by the optoelectronic frequency-modulated CW (FMCW)-THz-spectrometer "T-Sweeper"[37]. Offering continuous scan speeds up to 500 THz/s with a frequency coverage from 50 GHz to more than 4 THz, the T-Sweeper is typically used as real-time THz-spectrometer or photonic THz-RADAR. In contrast to its mostly used FMCW operation, we configure the T-Sweeper to emit a fixed frequency for our measurement. Two 90° off-axis parabolic mirrors in a U-configuration guide the emitted THz-signal from the source to the photoconductive receiver of the photonic spectrum analyzer. Figure 2 shows representative measured spectra at frequencies of 532 GHz (Fig. 2a) and 1.782 THz (Fig. 2b) recorded with a spectral resolution of 240 kHz. The

width of the spectral peak ranges between 31.8 ± 2.3 MHz at 532 GHz and 55.4 ± 4.0 MHz at 1.782 THz. The spectral peak ranges are a mixture of linewidth and frequency stability over the course of the measurement. They are about two orders of magnitude larger than the chosen resolution and several orders of magnitude larger than the spectral width of the beat note of the FC-referenced photonic LO which will be discussed further at a later point. Therefore, we confidently attribute the measured linewidth solely to the T-sweeper. We remark that the recorded linewidth and frequency stability of the T-sweeper operated at a fixed frequency may not reflect the performance parameters under fast sweeping. The peaks at both frequencies are well resolved with a peak power of −62 dBm and a noise power below −87 dBm at 532 GHz and a peak power of −75.5 dBm and a noise power of −78.5 dBm at 1.782 THz. The increase in noise power is the result of reduced responsivity within the photoconductive receiver in the PSA due to RC and lifetime roll-off effects[12,30]. In the PD transmitter, similar roll-off effects are present, leading to the reduction in signal power[31]. At 1.782 THz, the larger linewidth spreads the power of the THz source out more, further reducing the spectral power density recorded at each frequency point.

In order to showcase the spectral resolution of the photonic spectrum analyzer, a pulsed photonic emitter is used as source under test (SUT) in the second experiment, essentially the transmitter of a THz time-domain system. A modified, unreferenced TOPTICA difference FC (DFC Core+ 200) with an optical laser output power of 60 mW, a repetition rate of 200 MHz, and a central wavelength of 1,560 nm drives a Rh:InGaAs photoconductor[38] from the Fraunhofer Heinrich Hertz Institute, biased with up to 200 V. This pulsed THz-source generates a comb of THz-lines spaced by the repetition rate up to a frequency of at least 6.5 THz. Figure 3a) depicts the laser setup schematically.

Figure 3b) shows the spectrum of an individual line at a frequency of 300 GHz of the free running, only passively locked FC. It is measured by setting the spectrum analyzer's LO to a fixed frequency close to the SUT frequency, measuring the resulting IF signal in time and employing a fast-Fourier transformation. The combined linewidth is only 1.7 ± 1.3 Hz. As both laser systems are independent, the linewidth of 1.7 Hz is a convolution of the individual linewidths and thus represents an upper limit for both. Further, the pulsed laser is not actively locked so it can drift during the course of the measurement mimicking a larger linewidth. We therefore expect that the linewidth of the FC-referenced LO is smaller than that of the pulsed source. The phase noise of the pulsed THz source and of the LO scale about quadratically with the mode order, $n$[35,39]. Assuming an $n^2$ increase of the phase noise, the extrapolated phase noise from the combined linewidth of 1.7 Hz at 300 GHz remains below 1 kHz even at 6.5 THz. Such narrow linewidths

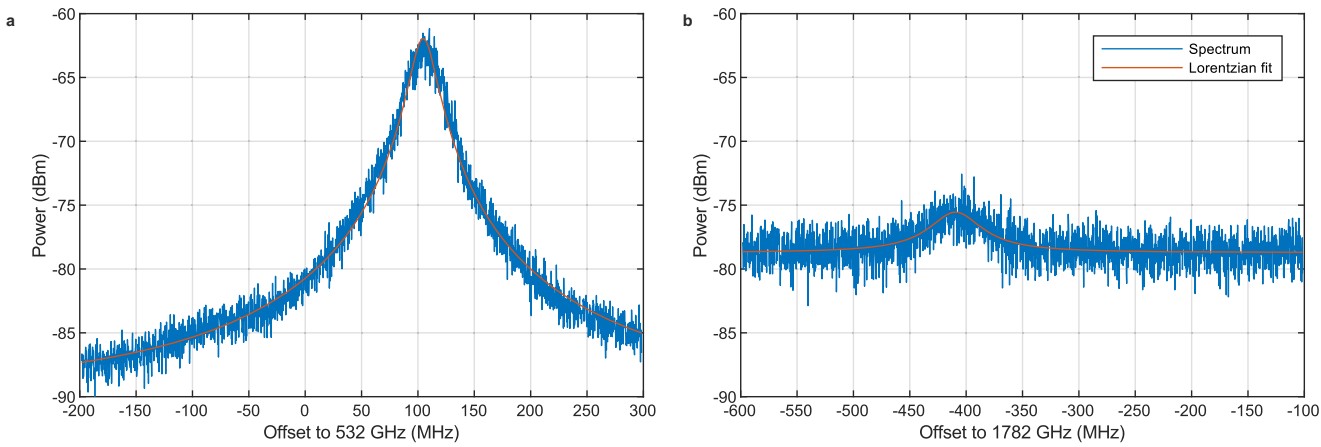

**Fig. 2 | Spectrum of a continuous wave THz signal. a** Emission spectrum of a CW-THz system (T-Sweeper) set to a fixed frequency of 532 GHz measured with a resolution bandwidth (RBW) and video bandwidth (VBW) of 240 kHz. **b** Emission spectrum of the same source at a fixed frequency of 1782 GHz.

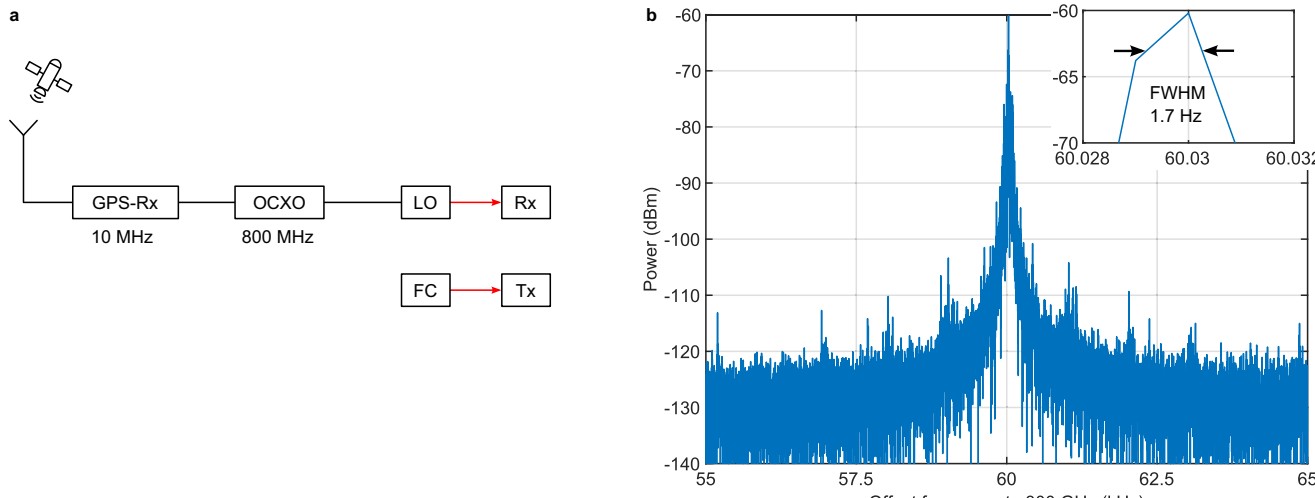

**Fig. 3 | Spectrum of unreferenced pulsed THz source. a** The FC is free-running during the spectrum analyzer measurements while the LO of the receiver (Rx) is referenced to GPS. **b** Acquired spectrum of the pulsed THz source at a frequency of 300 GHz with a resolution of 1 Hz. The inset shows the peak of the signal and the determined full-width half-maximum (FWHM) linewidth of 1.7 ± 1.3 Hz.

are in agreement with previous findings in ref. 40 where we determined a combined linewidth of 20 Hz at 339.6 GHz of a Menlo Systems time domain source characterized with an electro-optic CW comb that features a linewidth below 1 Hz.

The signal to noise ratio is as huge as 65.8 dB (Fig. 2b) at a measurement time of 1 s, despite that we investigate only a singular line of about 32,500 lines between DC and 6.5 THz emitted by the pulsed source at a total power of approximately 1 mW. The first and most important reason is the about 7 orders of magnitude smaller linewidth as compared to the previous case (Fig. 2), being similar for both systems and thus making maximum use of the equivalent noise bandwidth and system resolution. The peak is not smeared out over many frequency bins, all spectral power is concentrated in a 1.7 Hz wide narrow frequency window. Second, the frequency component under investigation at 300 GHz is close to the spectral maximum of the pulsed source between 500 GHz and 750 GHz. Above, the spectral power decreases exponentially towards higher frequencies. Third, the spectral response of the photoconductive receiver, characterized with a calibrated CW spectrometer, is lifetime and RC roll-off free below 570 GHz, offering maximum responsivity.

The spectral power is calibrated through determining the responsivity of the spectrum analyzer's receiver in a homodyne setup where the power is referenced to a calibrated THz power meter (for details see power calibration in methods section and calibration in supplemental material). The noise floor of the calibrated power meter limits the detectable frequency range to 1.6 THz. Relevant roll-off effects of the photoconductive mixer used as receiver are in effect above 1.1 THz enabling a responsivity estimation also for frequencies above 1.6 THz. With the responsivity information and the noise levels determined with the spectrum analyzer, we calculate the displayed averaged noise level (DANL) of the PSA (Fig. 4). Hereby, the DANL represents the noise level of the PSA for a resolution bandwidth (RBW) of 1 Hz. We reach a DANL of −145.6 dBm/Hz at a frequency of 100 GHz, −134 dBm/Hz at a frequency of 1 THz, and an extrapolated DANL of −105 dBm/Hz at 6.5 THz.

## kHz-level channel sounding with 4.75 THz coverage

Channel sounding is a technique of searching and identifying different channels for data communication[41]. Similar to a spectrum analyzer one may want to identify the absolute position, bandwidth and strength of a channel as well as undesired spurious emission into other channels. The frequency stability of the channels often benefit from referencing the signal to a GPS signal. GPS referencing also allows for comparable measurements of the same signal at different locations and times. This

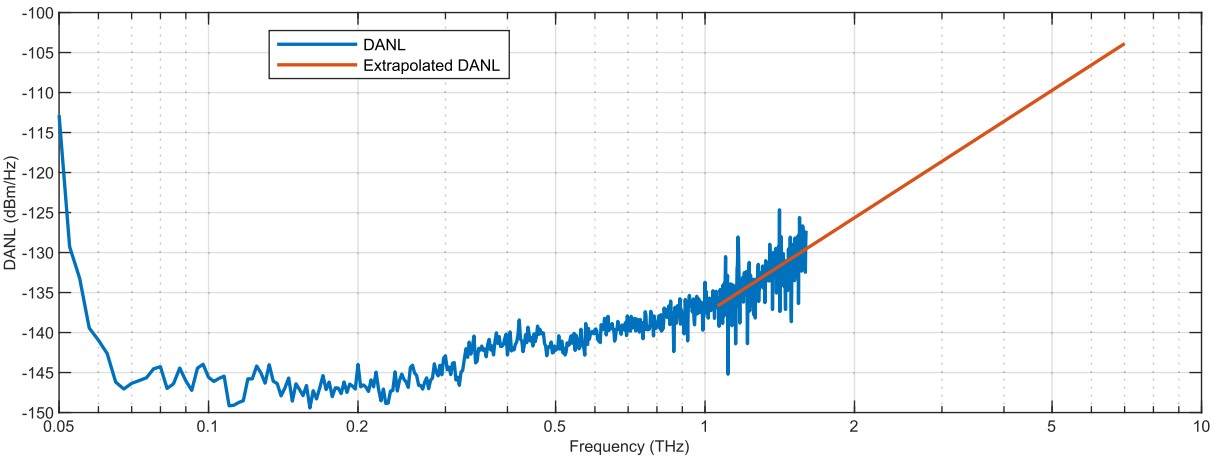

**Fig. 4 | Photonic spectrum analyzer noise floor.** Determined displayed average noise level (DANL) of the PSA for the frequency range between 50 GHz to 1.6 THz (blue curve). The orange curve is the extrapolation for frequencies up to 7 THz.

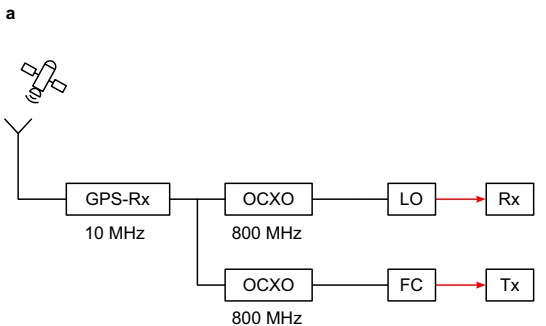

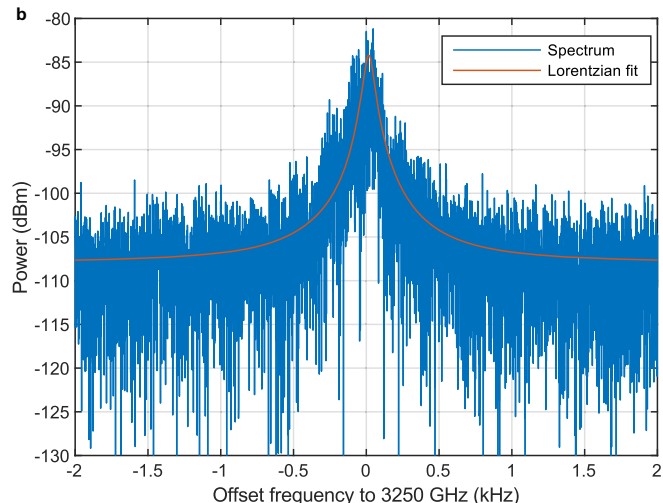

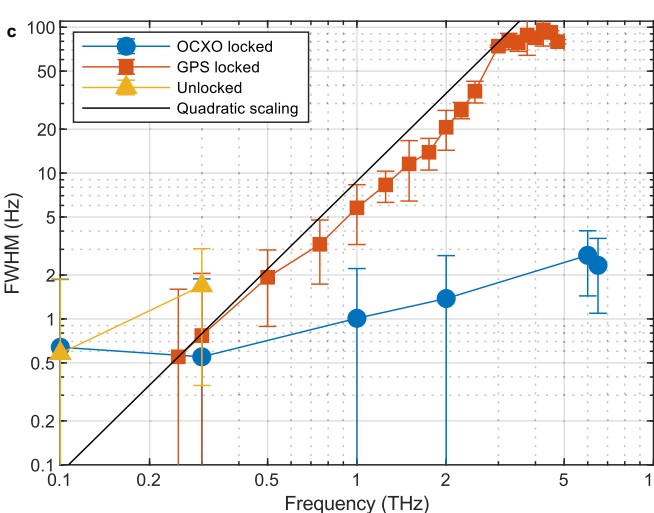

**Fig. 5 | Channel sounding. a** Spectrometer setup where the FC for the transmitter (Tx) and the LO are referenced to the same GPS signal each via its own OCXO. **b** Measured spectrum of the pulsed source at 3.25 THz with a RBW of 1 Hz and a measurement time of 1 s in the configuration shown in (**a**). The orange line shows the Lorentz-fit with a FWHM of 80.4 Hz. **c** Measured FWHM of the pulsed source using different referencing methods with standard measurement deviation. The yellow curve with the triangles shows the FWHM for the unreferenced pulsed source (c.f. Fig. 3a), the orange trace with squares for referencing to the same GPS signal (Fig. 5b) with the black line representing quadratic scaling. The blue curve with circles are the results of locking to the same OCXO (Fig. 6a).

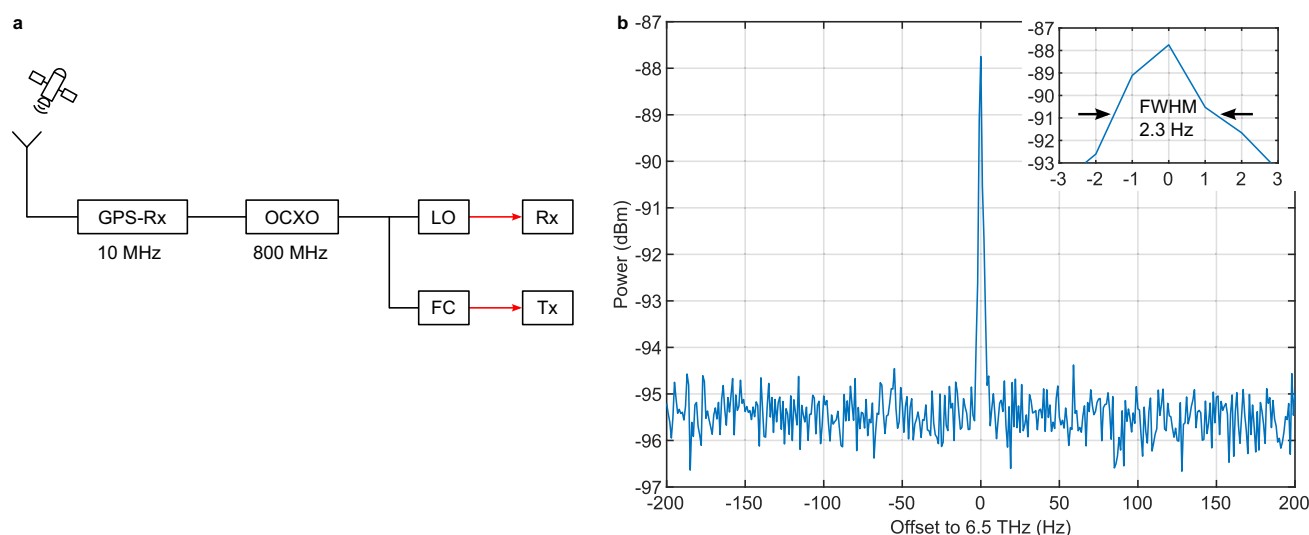

**Fig. 6 | Spectrometer. a** Spectrometer where both the FC and the photonic LO are referenced to the same OCXO. **b** Measured spectrum of the pulsed source at 6.5 THz in the configuration from (**a**). The data is averaged over 100 measurement runs. Each measurement run uses a RBW of 1 Hz and a measurement time of 1 s.

is experimentally simulated here by locking both the LO and the FC to the same 10 MHz GPS signal, yet each via its own OCXO (Fig. 5a). With the same THz setup as in the previous experiments, we measured several lines of the pulsed THz source.

Figure 5b) shows the measured spectrum of the pulsed source at 3.25 THz. The combined linewidth is Lorentzian with a width of 80.4 ± 10.4 Hz. Assuming purely quadratic scaling of phase noise with mode order, valid for independent FCs, the phase noise at 3.25 THz should be about 117 times higher compared to the previous 300 GHz measurement, resulting in an expected linewidth of 198 Hz. This is about twice the recorded value of 80.4 Hz. We therefore conclude that 50% of the phase noise is still uncorrelated, which stems from the additive phase noise of the optical phase locked loops, as well as the CW-laser noise outside the locking bandwidth. In other words, the CW-lasers can be considered to a large degree as independent entities despite locking to the same GPS signal. Figure 5c) (orange trace with squares) shows the development of the linewidth with respect to THz frequency for this scenario. It approximately follows a quadratic fit, in agreement with ref.[35]. The highest measurable frequency in this configuration was 4.75 THz. We remark that the noise floor at 3.25 THz in Fig. 5 b) of −107.8 dB (yet at an equivalent noise bandwidth of 1 Hz) shows the noise floor in direct vicinity to the signal, the actual noise floor at an offset frequency of 35 kHz to the signal is at −112 dB, i.e., another 4.2 dB lower. At a frequency resolution of 1 Hz this value is slightly larger than the estimated DANL of −117.2 dBm/Hz in Fig. 4. This is due to a combination of 1) the noise floor of the post detection electronics that play an increased role with the Fourier transformation mode used here and the frequency resolution of 1 Hz, 2) ground loop effects that influence the noise floor in our laboratory setup and 3) increasing absorbance and reflectance from the InP substrate[36].

### Spectrometer with 6.5 Terahertz frequency coverage

In order to measure the true signal shape, noise, and stability, a spectrum analyzer strictly requires that the LO and the signal under test are independent. In contrast, spectrometers used for investigating devices under test benefit from locking the LO with the signal under test. Channel sounding with locking to the same GPS signal already reduces the phase noise but for even better noise suppression both source and receiver can be locked to the same OCXO (Fig. 6a). Figure 6b) shows the spectrum of the pulsed source at a frequency of 6.5 THz with a combined (relative) linewidth of only 2.3 ± 1.2 Hz and a signal-to-noise ratio of at least 7 dB for 100 averages. Figure 5c (blue

line) also shows that the combined linewidth only slightly depends on the THz frequency and remains below 3 Hz over the whole frequency range. The linewidth corresponds to the remaining uncorrelated noise as well as drifts within the measurement duration irrespective of the linewidth of the individual lasers. The high correlation of both lasers drastically improves the signal-to-noise ratio, highly beneficial for spectroscopic applications. We estimate the power in the mode at 6.5 THz to 7 pW, which is now concentrated in a 2 Hz frequency bin as opposed to an 800 Hz linewidth in the non-referenced configuration, improving the signal to noise ratio by about a factor of 400. This improvement enabled the extension of the operation range very close to the current frequency limit of the photoconductive mixer, which originates from strong reflection and absorption of the Reststrahlenband of its Indium Phosphide substrate. The operation range from <50 GHz to 6.5 THz in a single setup is by far the largest frequency coverage of any THz system with sub-10 kHz resolution.

### Homodyne CW-THz-spectroscopy

Homodyne spectrometers profit from using the same LO signal for both the transmitter and the receiver, reducing common noise and enabling phase measurements. For this we replace the pulsed transmitter with a CW WIN-PD and excite it with the same LO as used for the receiver (Fig. 7a). A lock-in amplifier measures the photocurrent of the photomixing receiver while the LO sweeps the signal in frequency. Figure 7b presents the measured photocurrent of the receiver for frequencies between 100 GHz up to frequencies of 3.5 THz with frequency steps of 1.5625 MHz, acquired in a free-space reflection setup at ambient pressure with four off-axis parabolic mirrors. The measurements used an integration time of 50 μs for which the signal hits the noise floor at 3.6 THz. Longer averaging allows for accessing even higher frequencies[30,37]. The photocurrent decreases with the roll-off effects in the transmitter and receiver with increasing frequency as discussed earlier. Meanwhile distinct drops in the photocurrent are visible. These stem from absorption of the THz electrical field by water vapor present in the air and are based on rotational excitations of the molecules. The position and linewidth of the absorption lines are functions of the pressure and temperature (for details see method section).

In order to demonstrate the accuracy and resolution of the system, we investigate an almost Doppler-limited line of the water molecule by introducing a vacuum tube into the spectroscopy setup, adding a drop of ammonia water to the tube and evacuating to

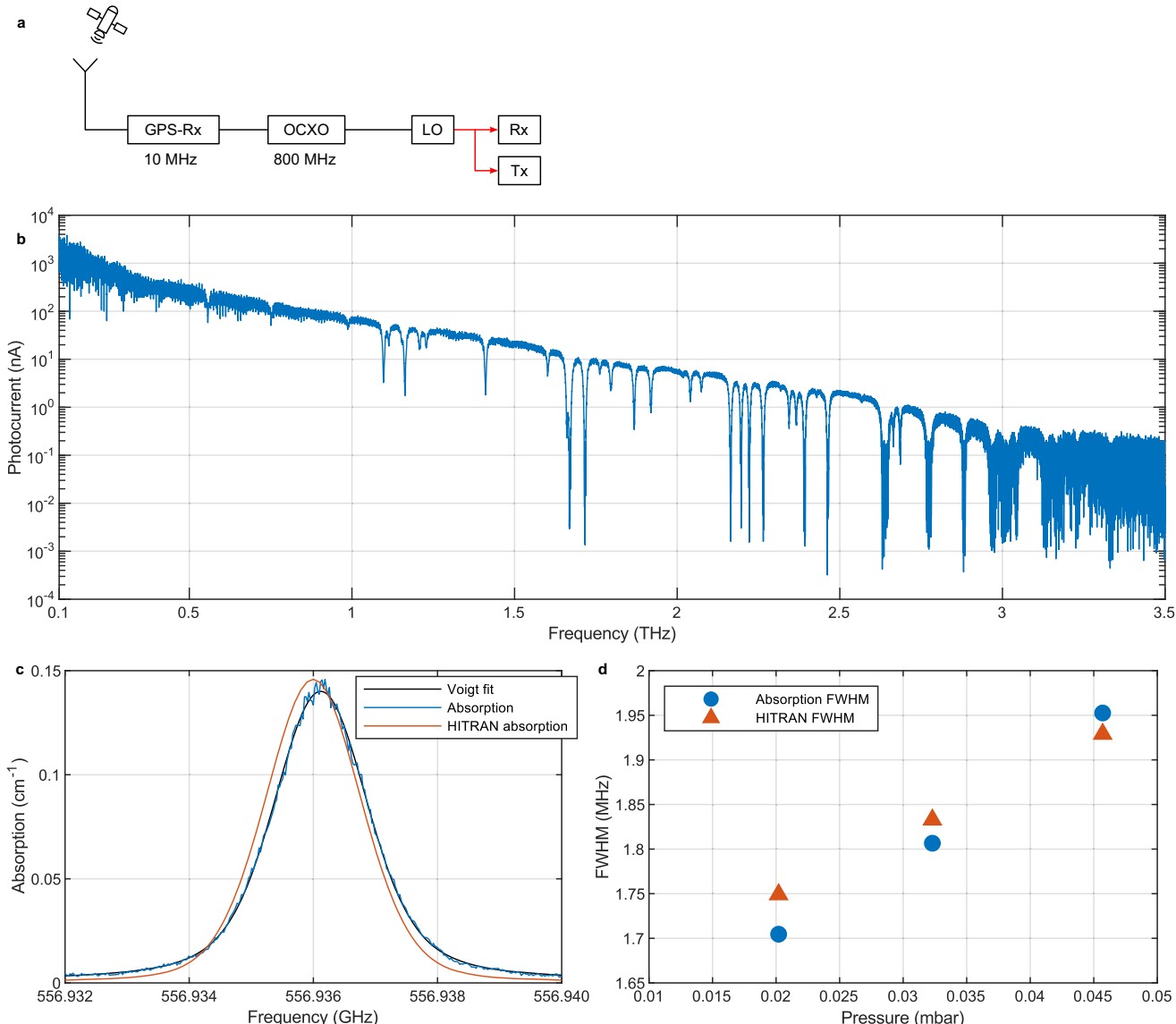

**Fig. 7 | Spectroscopy. a** In the homodyne THz-CW spectroscopy setup the LO excites both the transmitter and the receiver. **b** Measured photocurrent of the empty homodyne setup in reflection for frequencies between 100 GHz and 3.5 THz with an integration time of 50 µs and a step-size of 1.5625 MHz. **c** Absorption measurement (blue curve) and Voigt fit (black curve) with simulated absorption (orange curve) of water vapor at 0.032 mbar. **d** Measured (blue circles) and simulated (orange triangles) absorption full width half maximum (FWHM).

pressures between 0.020 mbar and 0.046 mbar (for more details on the measurement setup please consider the water vapour absorption spectroscopy section in the supplemental information). The absorption line of water vapor is close to 556.936 GHz (Fig. 7c) and has a linewidth of several GHz at atmospheric pressure and room temperature. Upon reducing the pressure, it becomes limited by the temperature broadened linewidth of 1.62 MHz[42] (see Eq. (8) in method section) caused by the optical Doppler effect of gas molecules at thermal velocity. The blue curve in Fig. 7c illustrates the measured water vapor absorption for a pressure of 0.032 mbar with a frequency resolution of 24 kHz, a measurement speed of 11.8 GHz/s, and averaged over 6 measurements. The black curve is a Voigt fit to the measurement data, and the orange line the calculated absorption from the HITRAN database[42] with a frequency resolution of 100 kHz, assuming a water vapor to air mixture of 25% ± 10%. The measured peak position is 120 kHz offset to the calculated peak position, corresponding to a relative deviation with respect to the HITRAN data of 215 ppb. The measured linewidth with 1.81 MHz is well resolved and only 1.1% smaller

than the linewidth of 1.83 MHz (Fig. 7d) calculated from HITRAN. Given an estimated error of 20% on the pressure measurements, these data confirm the accuracy and reference capability of the LO.

## Discussion

The demonstrated frequency coverage from microwaves to 6.5 THz is the largest so far for CW-THz spectrum analyzers and CW homodyne or heterodyne spectroscopy systems and was previously only accessible with purely pulsed time domain systems[38,43]. Yet, the latter only offer a resolution in the GHz range and therefore, at least a factor of $10^6$ worse than the resolution shown here. The state of the art tuning range in CW THz systems with photoconductors so far was 5.5 THz[30,31] achieved in a homodyne setup. Commercially available electronic spectrum analyzers only reach frequencies up to 1.5 THz. As Table 1 shows the presented photonic system significantly exceeds the frequency range of the electronic variants while achieving slightly better spectral resolution and comparable noise floors. Though frequency bands beyond 1.5 THz may be

**Table 1 | Spectrum analyzer performance comparison**

| | 100 GHz | | 1 THz | | 1.5 THz | | 6.5 THz (extrapolated) | |
|---|---|---|---|---|---|---|---|---|
| | Resolution [Hz] | Noise floor [dBm/Hz] | Resolution [Hz] | Noise floor [dBm/Hz] | Resolution [Hz] | Noise floor [dBm/Hz] | Resolution [Hz] | Noise floor [dBm/Hz] |
| ESA | 6 | −139 | 36 | −115 | 54 | −107 | NA | NA |
| PSA (this work) | 1 | −145.6 | 19 | −134 | 42 | −130 | 800 | −105.2 |

Typical specifications of commercial electronic spectrum analyzers (ESAs) for different waveguide bands taken from refs. 52,53 with a comparison to the photonic spectrum analyzer (PSA) introduced in this manuscript.

accessible in the future by electronic frequency extenders, fabrication and alignment tolerances as well as surface roughness of the metallic waveguides will increase losses, leading to higher noise floors i.e., less sensitivity. We remark that the emitted power of the investigated mode of the pulsed source at 6.5 THz is approximately 7 pW only. With a more powerful source at hand, such as a quantum cascade laser, e.g., a higher frequency coverage than the shown 6.5 THz will be achieved. Eventually, the Reststrahlenband of the InP substrate of the photoconductive mixer will limit the frequency coverage to about 7 THz. The Reststrahlenband is a frequency region which (nearly) totally reflects incoming radiation onto a polar dielectric due to a change in propagation of phonons within the material[44]. By removing the substrate the frequency coverage of the photoconductive mixer can be extended, as demonstrated in ref.[36], potentially within or possibly even beyond the Reststrahlenband of InP and InGaAs, to cover a similar range as the FC-referenced photonic LO. The FC-referenced photonic LO performs very similar in frequency stability and accuracy to other FC or cavity referenced photonic THz synthesizers[23,24]. Accounting for fiber thermal noise could further improve the performance[25]. Yet the system presented in this work combines the stability with by far the fastest tunability option of comparable synthesizers.

An alternative to the showcased homodyne system with the photonic LO at the transmitter and receiver side, the system combining the FC-referenced PSA and a phase-locked pulsed source (see section "kHz-level spectrometer with 6.5 THz frequency coverage" and Fig. 6) may be used. In its current state, the presented system can only step from comb line to comb line in 200 MHz steps with the current pulsed laser. However, the repetition rate of the FC can be tuned by up to 40 kHz. This causes a shift of a specific mode with its mode order $n$ by $n \cdot$ 40 kHz. This means that the free spectral range of 200 MHz will be completely covered by the mode at 1 THz and higher frequencies such that the system allows for gapless frequency coverage, suitable for kHz-level spectral resolution over the entire range from 1 THz to 6.5 THz with sufficient signal-to-noise ratio. The frequency range below, starting from approximately 50 GHz, is covered by the homodyne system. Therefore, the systems give access to low pressure trace gas detection of Doppler-limited lines over a large fraction of the THz range, which we showed exemplary by investigating the water vapor absorption line at 556.936 GHz. Terrestrial applications include breath gas analysis for detecting indicators for specific illnesses like acetone (diabetes, brain seizure), carbon monoxide (anemia, inflammation of respiratory system, smoke poisoning), methanol (nerval issues), ammonia (asthma, renal diseases)[45–47] or poisonous substances such as sulfur dioxide or carbonyl sulfide that may also be found in human breath[5]. A further terrestrial application is monitoring of environmentally dangerous gas species, e.g. carbon monoxide, sulfuric oxides, and nitrous oxides[48]. Further applications are, e.g., in astronomy where trace gases or even organic precursors in remote stellar Nebulas are detected with THz heterodyne receivers in order to identify molecules in star-forming regions. The source in this case would be the cosmic background radiation or heat generated in these regions, the receiver the PSA demonstrated in this manuscript. The extreme spectral resolution, frequency coverage and absolute

accuracy as a result of locking to GPS may further find applications in channel sounding of Terahertz communication systems[41], high precision spectroscopy of real and artificial materials, including functionalized resonators in bio sensors[49].

## Methods

### Photonic local oscillator

The laser engine, used as the LO, combines the stability and accuracy of comb technology with widely tunable CW external cavity diode-lasers (ECDL). The ECDL is optically phase-locked to a mode-locked erbium fiber laser comb with a fixed offset. For continuous phase-lock and tuning, the frequency comb is shifted with an external frequency shifter[32] based on serrodyne shifting with $2\pi$ phase wrapping with an electro-optic phase modulator. The tunable CW ECDL follows the applied frequency shift of the optical comb while inheriting the stability of the comb within the locking bandwidth. The CW laser covers a mode-hop free tuning wavelength range of 100 nm (> 12 THz) around 1550 nm within its and the comb's spectrum. The external frequency shifting simultaneously minimizes spurious signals in the vicinity of the relevant mode. Pairing the tunable CW laser with a fixed CW laser enables the generation of THz difference frequencies. In this case the phase noise of the difference frequency signal is given by the quadratically scaling phase noise of the optical comb with frequency[34] and the additive noise of the CW lasers. Additionally, combining the CW signals with a short pulse output of the frequency comb enables hybrid time- and frequency-domain measurements. Within the system all outputs are locked to the FC. Stabilizing the comb spectrum defines the noise properties. The comb has the option to absolutely reference it to a reference oscillator, e.g., an OCXO, which itself can be disciplined, e.g., to GPS, for SI traceable measurements. The referencing happens with a control circuit that matches the 4[th] harmonic of the repetition rate of the FC (200 MHz) to the frequency of the OCXO (800 MHz).

### Photonic mixer

The photonic mixer driven by the comb-referenced LO is a photoconductor optimized for operation at THz frequencies. Details on the used material Rh:InGaAs as well as typical antenna and electrode designs, can be found elsewhere[10,29,30]. In the following we only describe the working principle of the photoconductive continuous-wave LO in general. The two tones generated by the LO result in a beat note with a time-dependent laser power of

$$P_L(t) = P_{L,0}(1 + \cos[(\omega_1 - \omega_2)t + \varphi_1 - \varphi_2]) = P_{L,0}(1 + \cos[2\pi f_{LO}t + \varphi_{LO}]) \quad (1)$$

where we define the local oscillator frequency as the frequency difference of the two tones, $f_{LO} = |\omega_1 - \omega_2|/2\pi$. $\varphi_1$ and $\varphi_2$ are the phases of the optical carrier waves and $\varphi_{LO} = \varphi_1 - \varphi_2$ is the resulting envelope phase. The photoconductor absorbs the beat note generated by the LO, resulting in a conductivity modulation of $\sigma \sim P_L(t)$, by generating electron-hole pairs. The proportionality factor depends on a series of parameters, such as the absorption coefficient of the photoconductive layer stack but also the carrier lifetime and mobility of the material and

the electrode geometry. The antenna attached to the photoconductor receives at the same time the electric field $E_{SUT}(t)$ emitted by the source under test (SUT) and converts it to a voltage, resulting in a net current of

$$I_{IF}(t) \sim P_L(t)E_{SUT}(t) \tag{2}$$

The proportionality factor depends again on a variety of parameters, such as the antenna's radiation resistance, the RC roll-off caused by the combination of the antenna with the photoconductor's capacitance and a lifetime roll off. Generally speaking, the prefactor decreases with increasing frequency beyond a few 100 GHz.

For a delta-shaped, spectrally pure LO as approximately provided by the comb-referenced LO and a single frequency SUT with an electric field of $E_{SUT}(t) = E_{SUT,0} \cos(2\pi f_{SUT}t + \varphi_{SUT})$, the IF current (i.e. the frequency component closest to DC; THz components are ignored as they are filtered out by slow post detection electronics) becomes

$$I_{IF}(t) \sim P_{L,0}E_{SUT,0} \cos\left[2\pi(f_{LO} - f_{SUT})t + \varphi_{LO} - \varphi_{SUT}\right] \tag{3}$$

If SUT and LO are not phase-locked as in case of the spectrum analyzer, the phase difference $\varphi_{LO} - \varphi_{SUT}$ fluctuates randomly over time. Thus, the time-averaged current in Eq. (3) is zero. However, its power spectral density, $PSD_I$ remains finite, also for energy conservation reasons,

$$PSD_I(f, f_{LO} - f_{SUT}) = \left|\mathscr{F}\{I(t)\}\right|^2 \sim P_{L,0}^2 E_{SUT,0}^2 \delta(f - |f_{LO} - f_{SUT}|) \tag{4}$$

where $f_{IF} = |f_{LO} - f_{SUT}|$ is the intermediate frequency where the power spectral density appears and $\mathscr{F}(.)$ is the Fourier transformation. If the SUT features a finite bandwidth the process will map the THz bandwidth to the IF domain. In practice, the IF current in Eq. (3) generated by the photoconductor is pre-amplified by a transimpedance amplifier (TEM Messtechnik PDA-S or Femto DHPCA-100) and subsequently digitized by an analog-to digital conversion card (Advantech PCIE-1840L). The PSA in this manuscript employs two detection methods to determine the PSD of the SUT[22]. The first requires the LO to be set to one frequency during the course of the measurement. During the measurement time, the time-domain signal is acquired, Fourier transformed and finally squared, providing a "snapshot" of the spectrum around the LO frequency. To cover a larger frequency range, multiple of these are put together using slightly different LO frequencies. The second employs a (digital) low-pass filter in the IF-domain. For this the LO frequency needs to be swept from a starting to an end frequency with a known speed. Any part of the signal that falls into the low-pass filter will be acquired. Finally, squaring the signal provides the PSD-$E_{SUT,0}^2$ of the SUT. Alternatively, the PSD can be directly read out with a low frequency electronic spectrum analyzer operating in the IF domain[40]. The only missing step is a power calibration in order to determine the spectral power density of the received THz wave.

If Tx and Rx are phase-locked, as in the case of the CW spectrometer operated with the same photonic LO, the phase difference in Eq. (3) does not average out. Further, for the homodyne case $f_{LO} - f_{SUT} = 0$. Thus, the IF current becomes time-independent, $I_{IF} \sim P_{L,0}E_{SUT,0} \cos[\varphi_{Tx} - \varphi_{Rx}]$. Usually, there is a path length difference between the point where Tx laser signal and Rx laser signal are split and the position of the Rx. This optical path length difference $\Delta(nl)$ results in a linear frequency dependence of the phase difference, $\varphi_{Tx} - \varphi_{Rx} = k_0 \Delta(nl) = 2\pi c \Delta(nl)f_{LO} \sim f_{LO}$ oscillates with the THz frequency, causing homodyne fringes. Figure 7b plots the amplitude of the homodyne fringes vs. THz frequency.

## Linewidth and spectral resolution

For simplicity, we assume for now Gaussian spectral shapes of the form

$$E(f) = E_0 \exp\left(-\frac{(f - f_L)^2}{\sigma^2}\right) \tag{5}$$

where $\sigma$ is the $e^{-2}$ width of the spectral power ($e^{-1}$ width of the field) and $f_L$ the (positive-valued) laser frequency. The envelope of the heterodyned optical wave originates from a product of the fields at the respective colors in the time domain. Spectrally, the product turns into a convolution. For two Gaussian shapes of widths $\sigma_{L1}$ and $\sigma_{L2}$ results in a Gaussian with a combined linewidth of the photonic LO of $\sigma_{LO} = \sqrt{\sigma_{L1}^2 + \sigma_{L2}^2}$. Subsequently, the envelope with a spectral width of $\sigma_L$ is convoluted with the THz wave with a spectral width of $\sigma_{THz}$ leading to a combined linewidth of $\sigma_{tot} = \sqrt{\sigma_{LO}^2 + \sigma_{SUT}^2}$. Therefore, with the knowledge of the linewidth of either the SUT or the LO, the respective other one can be calculated. In any case, both are smaller than the measured linewidth $\sigma_{tot}$. For Lorentzian or Voigt spectral profiles, it is qualitatively similar, though mathematically more complicated.

## Power calibration

The power of the spectra is calibrated with the responsivity $\mathscr{R}$ of the Rh:InGaAs photoconductive receiver. The determination of the responsivity requires information on the resulting photocurrent $I_{IF}$ of a homodyne or heterodyne measurement setup with the photoconductive receiver and a measurement of the THz power $P_{cal}$ of the homodyne system. The responsivity is given as

$$\mathscr{R}(f) = \frac{I_{IF}^2(f)}{P_{cal}(f)} \tag{6}$$

Generally speaking, reliable power detection in the THz domain is very challenging and afflicted by large errors. For the power calibrations, we used a pyroelectric detector, calibrated to Si-units at the German national metrology institute PTB (SLT Sensor- und Lasertechnik, THz10). Due to differences in the calibration frequency of 1.4 THz and the measured frequencies, we assume a maximum frequency-dependent error of 30%. The pyroelectric detector has a noise equivalent power of 1 μW which enabled the calibration of the photoconductive receiver for the frequency range between 50 GHz and 1.6 THz. For higher frequencies, the source power of the calibration source is below the detection limit of the pyroelectric detector. Therefore, we extrapolated the responsivity for higher frequencies taking into account lifetime and RC roll-off of the photoconductor above 1.1 THz. To confirm the responsivity values, we additionally investigated a CW THz source at a frequency of 95 GHz with the PSA and measured its source power.

## Signal-to-noise ratio

The signal-to-noise ratio (SNR) within the measurements is dependent on the noise floor and responsivity of the receiver side, the THz power from the emitter and the linewidth of the signal-under-test and of the LO. All of the sources shown in this manuscript use photomixers (pin diodes or photoconductors) to generate the THz signal. Independent of the used optical signal (pulsed or CW), the power of the signal decreases with increasing frequency for frequencies higher than the roll-off 3dB-frequencies. For the pulsed variant, additionally, fewer optical modes mix with each other with increasing frequency, further reducing the signal power. The signal path consists in most cases of two parabolic mirrors in a U-configuration, Si lenses and InP substrate within the photomixer's package and a free-space path of less than 0.5 m. Along the signal path losses, reflection and absorption (i.e.,

water vapor, see also Fig. 7b) may occur reducing the signal strength inbound to the receiver. The photocurrent generated by the receiver depends on the strength of the incoming electrical field (see Eq. (2)) and the LO frequency. Roll-off effects within the photomixer reduce its responsivity with increasing LO frequency. The noise current of the photomixer is mostly independent of the LO frequency. The lower responsivities at higher frequencies thus require larger signals for successful detection. Additionally, the noise floor is directly proportional to the measurement bandwidth. Smaller measurement bandwidths (i.e., longer integration times) allow for the detection of smaller signals. The system-specific noise floor is therefore frequently given in the unit dB/Hz.

The bandwidth is limited on the lower end by the LO linewidth. The system essentially measures the spectral density, i.e., the power of the signal within a measurement bandwidth (frequency bin). Spectrally wide LOs or signals -such as the one in Fig. 2 are distributed over many frequency bins, reducing the power per bin and thus the signal to noise ratio. As the responsivity of the photoconductors drops with increasing frequency, smaller SNRs lead to smaller usable bandwidths, ranging from DC to the maximum still detectable signal. The result is a lower achievable measurement bandwidth in case of a non-referenced CW signal, while the usable measurement bandwidth increases with better relative stability between the SUT and LO enabling a measurement bandwidth of 6.5 THz with all spectral power concentrated within a 2 Hz frequency bin.

### Pressure and temperature broadening

The lifetime of the rotational state $\tau$ of a molecule limits its natural linewidth (typ. 10–100 s kHz) and features a Lorentzian spectral distribution $p(f)$,

$$p(f) = \frac{A}{(f - f_0)^2 + \left(\frac{1}{\pi\tau}\right)} \tag{7}$$

When two gas molecules collide or their dipole moments perturb each other during a pass by maneuver, the lifetime of the rotational state is lowered. The more molecules per volume are present (i.e. the higher the pressure) the shorter the respective lifetime and the larger the linewidth according to Eq. (7). The pressure broadened linewidth is directly proportional to the pressure, has a Lorentzian shape and reaches linewidths in the few GHz range at normal pressure[50]. We remark that heavier molecules have larger impact such that foreign gases will cause a different linewidth as a pure gas. In ambient conditions the absorption also experiences Doppler broadening caused by thermal motion of the gas molecules towards and away from the THz source. The Doppler broadening $\Delta f_D$ is given by ref. 51

$$\Delta f_D = f \sqrt{\frac{8\ln(2)k_B T}{mc^2}} \tag{8}$$

where $f$ is the center frequency, $k_B$ the Boltzmann constant, $T$ the temperature and $m$ the molecule's mass. Based on statistical motion of the molecules, Doppler broadening features a Gaussian line shape. The theoretical Doppler limit for water vapor is 1.62 MHz at the 556.936 GHz line at a room temperature of 23 °C.

For the measurements with water vapor under vacuum, we used a spectroscopy setup with a vacuum tube in the THz beam. Beamforming equipment in the form of two lenses and photoconductive transmitter and receiver were outside the vacuum tube, purged with dry air to prevent water absorption at normal pressure outside the tube. A combined membrane pump with a turbomolecular pump enabled pressures as low as 0.001 mbar within the tube. Additional water in form of an ammonia-water solution is added in an extended part of the tube that is separated by a needle valve. For the measurements the needle valve is opened slightly to let water vapor into the

tube and depending on the pressure the pumps closed off of the tube. Each measurement consists of three runs of five seconds where each run contains one up and one down scan. During each measurement we monitored the pressure with a pressure gauge. The pressure within the tube changed on average by a factor of 1.5 as the tube is quite leaky. Therefore, each measurement is compared to the maximum pressure within its measurement time window. The pressure gauge has an error of 20% due to a lack of recent calibration. The tube does not contain heating or cooling equipment and was left at room temperature (23 °C), limiting the measurements to the Doppler broadened spectra. The absorption coefficient $\alpha$ is calculated by

$$\alpha = -\frac{\ln\left(\frac{I_{H_2O}}{I_0}\right)}{d} \tag{9}$$

where $I_0$ is the intensity of the reference measurement, $I_{H_2O}$ the water vapor measurement and $d = 33$ cm the mean THz path length inside the vacuum tube.

## Data availability

All the data supporting the findings have been placed in the repository of the Technical University of Darmstadt, Darmstadt, Germany. The data are under restricted access due to legal matter; access can be obtained by contacting either of the corresponding authors via E-mail.

## Code availability

No custom code was used for statistical analysis. Code for evaluating the data is available in combination with the data under the above-mentioned conditions. Any other code, i.e., for data acquisition or device controlling, are not publicly available due to legal matters.

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

## Acknowledgements

This research is financially supported by the European Research Council (ERC) through the Proof of Concept Grants "PhoSTer-THz", grant agreement number 101057162, and "PhoVeNA", grant agreement number 101149200.

## Author contributions

B.K. set up the software and performed the experiment together with S.M., T.P., and L.L.; B.K., S.M., and T.P. conducted the ammonia absorption measurements; S.M., T.P., R.W., and N.V. designed and were responsible for the comb-referenced L.O.; S.P. conceived the basic idea of the photonic spectrum analyzer and the experiment; L.L., M.D., and R.K. developed the photomixers and photodiodes employed, and G.S. conducted the CW THz spectroscopy measurements. The manuscript was mainly written by S.P. and B.K. and revised by all authors.

## Funding

## Competing interests

The authors declare no competing interests.
