## [Transparent Peer Review file · Nature Communications]

Terahertz photonic heterodyne spectral analysis with (sub-) kHz resolution and 6.5 THz frequency coverage

Corresponding Author: Mr Benedikt Krause

Version 0:

Reviewer comments:

Reviewer #1

(Remarks to the Author)

The authors present a comb-based THz spectrometer devised with a photonic local oscillator (LO) generating reference THz waves by photo-mixing a pair of diode lasers (one is fixed, the other tuned) with phase locking to a commercial comb (DFC CORE+, Toptica). THz signal detection is made using Rh:InGaAs photoconductive antenna (PCA) via a transimpedance amplifier. A series of hetero- and homodyne tests are demonstrated over 100 GHz to 6.5 THz range. The authors claim that their comb-based setup outperforms state-of-the-art electronic systems in terms of frequency coverage and system cost with a competitive noise floor and spectral resolution.

The concept of comb-based THz spectroscopy over a large spectral range is well established in the community. Additionally, the generation of low-noise THz waves by phase-locking a pair of diode lasers to a stabilized comb has been well demonstrated. In this context, the authors' approach is not considered particularly novel, and the achieved level of spectral resolution is not outstanding compared to recent records in the literature. Furthermore, the manuscript requires more technical details on the experimental setups and test results. Therefore, no favorable recommendation is given for the manuscript to be accepted or considered for publication in Nature Communications.

Reviewer #2

(Remarks to the Author)

The authors introduce a wideband photonic analyzer that supports both heterodyne and homodyne detection over an impressive part of the Terahertz domain with a sub-kHz resolution. It may be used either as a spectrum analyzer to evaluate the spectral shape and the spectral stability of a source under test or as a tool to perform comb-calibrated optical spectroscopy over an unparalleled range from below 100 GHz to 6.5 THz. The analyzer condenses technical approaches that were already demonstrated by the authors, namely the comb-locked frequency swept synthesis developed by Toptica (Refs. 28-30) and the generation of tunable THz radiation by laser photo-mixing demonstrated by the Fraunhofer Institute for Telecommunications (Ref. 32). Nevertheless, since this combination is shown to provide a paradigmatic shift in the synthesis and in the analysis of THz signals, I recommend the publication of this paper provided that the authors address two minor revisions and one remark.

- I would better emphasize, perhaps with an experimental example, the metrological attitude of the spectrometer in the THz domain, which naturally derives from the comb-calibrated frequency scale.

- I would better comment the poor signal-to-noise ratios shown in Fig. 2b and 5b. This seems to conflict with the good noise figures shown in Table I.

Remark: there is an intensive advertisement in Figs. 1, 3, 5, 6. of the frequency comb source produced by Toptica, namely the DFC core: name and model of the used comb source should definitely appear just once in the text (and not in the figures). This is important also because the paper already enforces Toptica's proprietary technology on the comb-calibrated frequency-swept synthesizer. The paper should remain agnostic, at least to a reasonable level, with respect to industrial interests.

Reviewer #3

(Remarks to the Author)

The work reported by Krause et al. demonstrates a THz spectrum analyzer with high resolution which covers a broad range of frequencies. Their concept takes advantage of a frequency comb to create two pure optical tones from two diode lasers, one at a fixed frequency and the other with a tunable frequency. By heterodyning these two optical tones, the authors modulate at a precise LO frequency the conductivity of a photoconductive antenna that is able to record THz from an external source. As the external THz signal mixes with the tunable LO frequency, low noise electronic can be used to record the spectrum of the external THz source with a precision defined by the excellent spectral purity of the comb-reference LO. The authors also demonstrate that their photoantenna allows the detection of frequencies up to 6.5 THz. The authors demonstrate the performance of their system in 4 experiments, which are the last 4 sub-sections of the Results section. They use different THz sources and configuration while reporting on the spectral resolution, noise and spectral range of their receiver.

The manuscript contains findings that will be of interest to experimentalists working in the field of THz source characterization. However, the authors do not explicitly state the novelty aspects allowing their technique to achieve kHz-level resolution over a spectral band ranging from 0.1 to 6.5 THz. In fact, the technique seems to overlap significantly with some of the previous literature. As mentioned by the authors, the concept of using photonic heterodyne spectrum analyzers for THz detection has already been established in Ref [22-25]. Furthermore, details about the broadband Rh:InGaAs photoconductive receiver was already reported in Ref [26]. Finally the concept referencing a CW laser to a comb was reported as early as 2021 by some of the same authors in Ref [28].

Comments:

I find very interesting the concept of referencing the LO to an Oven Controlled Crystal (Xtal) Oscillator (OCXO), which is itself referenced to a 10 MHz reference signal locked to GPS. This discussion and the implication of this scheme are a bit hidden in the current version of the manuscript. I think they could be emphasized to amplify the impact of this work.

The authors claim that their receiver is capable of resolving frequencies up 10 THz. It's not clear if this is possible though, since it has not been demonstrated in this work. Carrier lifetime in the photoconductor material could notably impose a fundamental spectral limit. The authors also mention that "the Reststrahlenband of the InP substrate of the photoconductive mixer will limit the frequency coverage to about 7 THz". This might also be a fundamental limit as it is not clear how the receive would perform without this substrate. Therefore, the authors must refer to a maximum frequency detection reaching 6.5 THz (which is demonstrated in this paper) instead of 10 THz. The authors must also change the wording in their abstract which also seems to indicate that their receiver operate up to 10 THz: " We present an ultra-wideband, continuous-wave photonic receiver with kHz-level spectral resolution in the terahertz domain (100 GHz-10 THz)".

The paper mentions some limitations of their receiver due to the spectral fluence of the source, e.g. at Line 149. A more in-depth discussion would be necessary to determine how the fluence of the source may limit the performances of their receiver.

In the section "Spectrum analyzer for unreferenced sources", the authors should show the full spectrum from 0.1 to 6.5 THz in addition to the resolution of a single comb line in Fig. 3b.

The authors have showed homodyne CW THz-spectroscopy in the paper with an integration time of 50 μ s where the signal is distinguishable until 3.5 THz. One of their previous articles show a spectrum reaching 5 THz with an integration time of 300 ms (Ref [26]). I recommend using 300 ms here also to allow for a better comparison between this configuration and the previous one used in Ref. 26.

Please check: the abbreviation of oven-controlled crystal (xtal) oscillator should be OCXO, not OXCO.

Regarding the sentence: "While a spectrum analyzer strictly requires that the LO and the signal under test are independent in order to measure the true signal shape, noise and stability, this independence is undesired in spectrometers.", the authors should provide more information to explain why they are making these assumptions. It is advantageous (but maybe not necessary) that a spectrum analyzer is independent from the signal it analyzes, but it is less clear why this independence is "undesired" for spectrometers. Most spectrometers don't have any "dependence" with the source they characterize.

This sentence: "Fig. 6b) shows that locking to a common reference allows to extend to a maximum frequency of 6.5 THz", it's not clear. First, Fig. 6b only shows a power signal, not a comparison for locked and unlocked signals. Also, I believe that the authors are making a confusing shortcut: using the same OCXO probably leads to an improved signal-to-noise, this in turns might lead to an increase in the detection spectral range. This should be clarified.

The sentence "Key applications are future high-speed wireless communication, with several THz bands already defined in 6G [1]" could be misleading because the Ref [1] mentions that frequencies between 1 THz and 3 THz have not yet been defined as this spectral region can be used by both active and passive services. Plus, nothing is mentioned in Ref [1] about frequencies above 3 THz.

This sentence: "For the highly important field of THz wireless communication an opto-electronic device is even a prerequisite to transform the optical signal, transmitted through a fiber, to a wireless THz signal." should be removed since the statement is not always true. For example, a nonlinear medium could be used to transform the optical signal into a wireless THz signal.

This sentence could be misleading: "The extenders double or triple the microwave baseband frequency several times up to the desired THz frequency or from the THz frequency down to the base band [15]." Ref 15 demonstrates this concept up to 400 GHz and intrinsic carrier response time in materials still limit these applications to about 1 THz only.

There's no verb in this sentence: "For example, a Hilbert-transform 66 spectrum analyzer based on a high-Tc Josephson junction [17] with a high scanning speed of up to 4 67 THz/s and a frequency coverage of at least 1 THz, yet only reaching a resolution of 50 MHz."

This sentence is long and confusing: "These utilize continuous-wave (CW) lasers as local oscillator (LO) – namely, an electro-optical comb based on an electro-optical phase modulator at telecom wavelengths with a frequency coverage of up to 1 THz and a resolution of 1 Hz [22], free-running distributed feedback (DFB) laser diodes with a difference frequency coverage of up to 4.5 THz at a wavelength of 780 nm [25] and a resolution of 1.2 MHz at a center wavelength around 1550 nm [23] or DFB laser diodes at 850 nm referenced to an electronic THz signal with a frequency coverage of 1 THz and a resolution of 100 Hz [24]."

Reviewer #4

(Remarks to the Author)

Version 1:

Reviewer comments:

Reviewer #1

(Remarks to the Author)

The authors' work is very much related to the paper "Photonic comb-rooted synthesis of ultra-stable terahertz frequencies" published by DC Shin, BS Kim, H Jang, YJ Kim, SW Kim in Nature Communications (2023)14:790. Nonetheless, the 2023 Nature Comms paper was neither cited nor discussed even in the authors' revised manuscript -- It is strongly suggested that the authors describe how their work is differentiated and valued comparatively in terms of the THz generation system setup, frequency instability, phase noise and tuneable range before making final claims on their achieved performance.

Reviewer #2

(Remarks to the Author)

The authors have thoroughly addressed all reviewers' concerns and well complemented the paper with missing measurements, clarifications, extensions. I am in favor of the publication of the manuscript in Nat Comm.

Reviewer #3

(Remarks to the Author)

I appreciate the work invested by the authors to address my comments as well as the comments from the other Reviewers.

-I am still not convinced, based on experimental evidence, that the performance beyond 10 THz is achievable. This was my "concern #2". The authors answered: "[...] some of the authors have demonstrated photoconductor performance beyond 10 THz with devices where the substrate (and hence absorption by its Reststrahlenband) was essentially removed [39]". I couldn't find this photoconductor performance demonstration beyond 10 THz in reference [39], which is Gordon et al. J. Quant. Spectrosc. Radiat. Transfer 277, 107949 (2022). The authors also write in the text: "Removing the substrate allows for coverage beyond 10 THz [33]." But reference [33], which is OE 20, 23896 (2022), doesn't show (or even claim) to experimentally demonstrate THz detection beyond 10 THz. In that previous work, the measured signals are very close to the noise level after 9.5 THz. Because of the results presented in this paper and in previous work referenced by the authors [33, 39], I recommend removing any claims on the performance of these devices exceeding 10 THz when the substrate is removed.

-The authors propose a plausible cause to explain the slightly larger spectral linewidth of ammonia, which they attribute to the presence of contaminating water molecules. However, the THz spectrum should allow them to test directly this hypothesis by monitoring spectral components at water molecule absorption peaks. The authors should employ this technique to confirm their hypothesis or provide a justified explanation for any limitations preventing its implementation.

-The authors should remove the subjective word "excellent" in the added sentence: "By simply sweeping the LO frequency, we can record the spectrum of any unknown source under test at a resolution defined by the excellent spectral purity of the FC-referenced LO."

Version 2:

Reviewer comments:

Reviewer #1

(Remarks to the Author)

The authors have addressed the reviewer's concern by adding other relevant LO / THz frequency synthesizer methods to the introduction with a short comparison in the revised manuscript. Now this reviewer agrees to accept the manuscript for publication.

Reviewer #3

(Remarks to the Author)

I appreciate the effort the authors have made to address the reviewers' comments, and I believe this manuscript merits publication in Nature Communications. That said, I would still encourage the authors to reconsider the claim that the presented THz detection technology can operate effectively beyond 10 THz. As it stands, this assertion is not directly supported by the data in the manuscript, and relies solely on a single conference paper: Proceedings Volume 12885, Terahertz, RF, Millimeter, and Submillimeter-Wave Technology and Applications XVII; 1288503 (2024)

While figure 3b from that conference paper does show a THz signal above the noise floor extending to 10 THz, the nature of this signal is ambiguous. Since the spectrum reflects the absolute value of the Fourier-transformed time-domain signal, the high-frequency components, 6 to 7 orders of magnitude weaker than the low-frequency signal, could plausibly arise from minor amplitude fluctuations of a time-resolved signal containing only low frequencies. This contribution is distinct from the "noise" indicated in the figure. In previous experiments, such high-frequency artifacts could be entirely suppressed by applying a spectral filter (high pass) that reduces the low-frequency contribution of the THz signal (e.g. see Fig. 2a in Light Sci Appl 14, 44 (2025))

This is why I find the repeated claim of >10 THz operation mildly problematic. Although not a cause for rejection, reiterating this claim in a Nature Communications article risks prematurely solidifying the notion that this device has been demonstrated to operate in that regime. I am not suggesting the claim is incorrect, only that the supporting evidence is very weak. The concern is that future researchers may cite this work as a definitive proof of >10 THz operation, potentially motivating follow-on studies based on an unverified premise.

Response to reviewers on Nature Communications manuscript NCOMMS-24-72395-T

Dear Reviewers,

We sincerely thank you for the comments and suggestions on the manuscript. Please find below a detailed response to each of the remarks individually. We excuse for the long duration of the revision. It was due to the necessity of taking new data, yet requiring to assemble a new gas spectroscopy setup. We hope to have improved the manuscript to allow for further consideration.

Yours sincerely,

Benedikt Krause

(Corresponding author, in the name of all authors)

Reviewer #1, Concern #1: The authors' approach is not particularly novel and the achieved level of spectral resolution is not outstanding compared to recent records in the literature.

Author response: To address this concern of the reviewer, citations for the statement would be helpful. Comparing the current work to state-of-the-art, including the references given in the paper, we would like to point to the following results that significantly exceed previous work:

- 1.) Frequency coverage: We achieved a frequency coverage from < 100 GHz to 6.5 THz under continuous-wave operation with semiconductor photomixing devices, excelling the state-of-the-art by 1 THz (5.5 THz in Refs. 26 and 27). Such frequency coverage was only accessible to date with pulsed time domain systems that offer about a million times lower spectral resolution.
- 2.) Tuning speed and no ambiguity: The system is easily tunable over the whole range in a plug & play configuration. When compared to comb-based pulsed systems, the presented system is not limited to individual comb lines, there is also no ambiguity which mixing product of which comb lines is currently being detected. Instead, the photonic LO system, ALTAS, is easily and extremely quickly (up to ~ 1 THz/s) tunable and is truly CW.
- 3.) Low noise floor: We achieve a displayed average noise level/noise floor better than -145 dBm/Hz (Fig. 4; corresponding to 3 aW for a 1 Hz noise bandwidth) between 100 GHz and 200 GHz and still -134 dBm/Hz, corresponding to 40 aW for 1Hz noise bandwidth at 1 THz. These values are by far the lowest noise floors for photonic receivers in the literature and are at the same level as the best electronic spectrum analyzers.

Further, to the best knowledge of the authors, the manuscript is the first to combine a pulsed photonic THz source with a CW photonic THz receiver for spectrometer measurements. Simultaneously, the LO shows a competitive linewidth with 1.7 Hz at 300 GHz and less than 1 kHz at 6 THz with the option to lock the LO to GPS allowing for referenceable measurements basically anywhere on earth with expected similar performance. We are aware that there are demonstrated combs with (much) lower linewidth, but they needed complicated and expensive locking to frequency standards, e.g. hydrogen masers or atomic clocks.

For comparison within the scope of Nature journal publications: we found recently published papers with similar linewidths but considerably lower frequency coverage and considerably higher noise floors: Nature Photonics 18, 1230-1231 (2024); Nature Communications 14, 7162 (2023).

For further improvements to our contribution with focus on metrological applications please also see the responses to reviewer #2, concern #1 and reviewer #3, concern #1. We conclude that the system presented in our paper outperforms the state of the art in many ways, in particular in terms of frequency coverage and noise floor.

Author action: We incorporated measurements on the absorption spectra of an ammonia solution under vacuum to compare the accuracy of the LO to nature.

"In order to demonstrate SI unit tracability and resolution of the system, we investigate pressure broadening of the ammonia molecule by introducing a vacuum tube into the spectroscopy setup, adding a 25% ammonia solution to the tube and evacuating to pressures between 0.018 mbar and 20.6 mbar (for more details on the measurement setup please consider the supplemental information). The absorption line of ammonia is close to 572.5 GHz (Fig. 7c) and has a linewidth of several GHz at atmospheric pressure and room temperature. Upon reducing the pressure it becomes limited by the temperature broadened linewidth of 1.72 MHz for pressures below 0.1 mbar [39] (Fig. 7e, orange curve). The blue curve in Fig. 7c illustrates the measured absorbance for a pressure of 0.041 mbar with a frequency resolution of 24 kHz and averaged over 6 measurements. The black curve is a Voigt fit to the measurement data and the orange line the calculated absorption from the HITRAN

database [39] with a frequency resolution of 100 kHz. The measured peak position is 80 kHz offset to the calculated HITRAN peak position confirming the accuracy and reference capability of the LO. The measured linewidth with 2.4 MHz is well resolved and slightly larger than the theoretical Doppler limited linewidth of 1.72 MHz which may be attributed to foreign broadening caused by water molecules present from the ammonia solution. With increasing pressure in the tube, i.e. at 16 mbar (Fig. 7d), the measured linewidth increases in good agreement with the calculated absorption (Fig. 7e) yet faces an up-shift for the FWHM resulting from the foreign broadening. Fixing the Gaussian component of the Voigt fit, as we do not expect a temperature change, the measured Lorentzian part, responsible for the pressure broadening, (represented by Γ in Fig. 7f) increases similar to the HITRAN-calculated value with a similar upshift as the FWHM.” (lines 298-317)

Reviewer #1, Concern #2: The manuscript requires more technical details on the experimental setups and test results.

Author response: We are happy to introduce more technical content. For further details, please also see reviewer #2, concern #2, reviewer #3, concerns #1 and #3.

Author action: We added further details within the manuscript, the method section and the supplementals and believe to have captured all important information. In case we miss important information please let us know which ones, we are happy to include them.

“Within all measurements shown in this manuscript the LO is additionally referenced via an oven-controlled crystal oscillator (OCXO) to a global positioning system (GPS) signal. The OCXO generates an 800 MHz reference signal that phase-locks to the 4th harmonic of the repetition rate of the FC. The additional referencing of the OCXO to the GPS gives the LO excellent absolute frequency accuracy which we verify in a later section.” (lines 131-135)

“In practice, the IF current in Eq. (3) generated by the photoconductor is pre-amplified by a transimpedance amplifier (TEM Messtechnik PDA-S or Femto DHPA-100) and subsequently digitized by an analog-to digital conversion card (Advantech PCIE-1840L). The PSA in this manuscript employs two detection methods to determine the PSD of the SUT [22]. The first requires the LO to be set to one frequency during the course of the measurement. During the measurement time, the time-domain signal is acquired, Fourier transformed and finally squared leaving us with a “snapshot” of the spectrum around the LO frequency. To cover a larger frequency area, multiple of these are put together using slightly different LO frequencies. The second employs a (digital) low-pass filter in the IF-domain. For this the LO frequency needs to be swept from a starting to an end frequency with a known speed.” (lines 424-433)

“For the measurements with the ammonia solution under vacuum we used a spectroscopy setup with a vacuum tube in the THz beam. Beamforming equipment in the form of two lenses and photoconductive transmitter and receiver were outside the vacuum tube. A combined membrane pump with a turbomolecular pump enabled pressures as low as 0.001 mbar within the tube. The ammonia solution is added in an extended part of the tube that is separated by a needle valve. For the measurements the needle valve is opened slightly to let ammonia vapor into the tube and depending on the pressure the pumps closed off of the tube. Each measurement consists of three runs of five seconds where each run contains one up and one down scan. During each measurement we monitored the pressure with a pressure gauge. The pressure within the tube changed on average by a factor of 1.5 as the tube is quite leaky. Therefore, each measurement is compared to the average pressure within its measurement time window. The tube does not contain heating or cooling equipment and was left at room temperature (23 °C) limiting the measurements to the Doppler broadened spectra.” (lines 521-532)

“For the calibration of the heterodyne measurements, we employ the responsivity (see eq. (6)) acquired from homodyne measurements with the employed Rh:InGaAs photoconductive receiver. The homodyne system employs the same local oscillator signal for both the transmitter and receiver. The transmitter is a waveguide-integrated photodiode provided by the Fraunhofer Heinrich-Hertz-Institute (HHI) and the local oscillator consists of a Keysight 81960A at 1540 nm, along with a tunable Keysight 81608A at longer wavelengths. Their difference frequency is the local oscillator frequency. Both the transmitter and the receiver are driven by an optical power of 30 mW each. A lock-in amplifier modulates the transmitter bias with a sinusoidal shape that also provides the reference for demodulation of the received signal. This measurement determines the photocurrent I_R of the photoconductive receiver in response to a given electrical field on the input side. Following the responsivity equation (Eq. 6 in the methods section), the photocurrent is squared and the power of

the calibration source determined by a gauged pyroelectric detector (SLT Sensor- und Messtechnik THz10). This pyroelectric detector is calibrated at a frequency of 1.4 THz at the German National Metrology Institute PTB and possesses a noise floor of 1 μ W." (Supplementals lines 5-18)

Reviewer #2, concern #1: I would better emphasize, perhaps with an experimental example, the metrological attitude of the spectrometer in the THz domain, which naturally derives from the comb-calibrated frequency scale.

Author response: We thank the reviewer for the suggestion. Due to the direct locking of the LO to GPS, the system should have an excellent absolute frequency accuracy. We investigate this by measuring the absorbance of an ammonia solution at 572.5 GHz at pressures between 0.018 mbar and 20.6 mbar. As we cannot control the temperature within the vacuum tube, we are limited by the Doppler broadening which is 1.71 MHz at room temperature for ammonia. The acquired data has a selected frequency resolution of 24 kHz achieving good agreement to calculated data based on the HITRAN database.

Author action: We incorporated absorption measurements of an ammonia solution under vacuum into the section “homodyne CW-THz-Spectroscopy” and moved the silicon wafer reflectance measurement to the supplemental material.

“Locking the photonic system to GPS enables tracing back the measured parameters to SI units, being of key importance for metrological applications.” (lines 23-25)

“Here we demonstrate a concept that combines the capability for heterodyne spectrum analysis and high-resolution spectroscopy up to 6.5 THz in a single measurement instrument while referencing the system to GPS for a high absolute frequency accuracy.” (lines 90-92)

“GPS referencing also allows for comparable measurements of the same signal at different locations and times.” (lines 228-230)

“HOMODYNE CW-THz-SPECTROSCOPY

Homodyne spectrometers profit from using the same LO signal for both the transmitter and the receiver, reducing common noise and enabling phase measurements. For this we replace the pulsed transmitter with a CW WIN-PD and excite it with the same LO as used for the receiver (Fig. 7a). A lock-in amplifier measures the photocurrent of the photomixing receiver while the LO sweeps the signal in frequency. Fig. 7b) presents the measured photocurrent of the receiver for frequencies between 100 GHz up to frequencies of 3.5 THz with frequency steps of 1.5625 MHz, acquired in a free-space reflection setup at ambient pressure with four off-axis parabolic mirrors. The measurements used an integration time of 50 μ s for which the signal hits the noise floor at 3.6 THz. Longer averaging allows for accessing even higher frequencies [27, 34]. The photocurrent decreases with the roll-off effects in the transmitter and receiver with increasing frequency as discussed earlier. Meanwhile distinct drops in the photocurrent are visible. These stem from absorption of the THz electrical field by water vapor present in the air and are based on rotational excitations of the molecules. The position and linewidth of the absorption lines are functions of the pressure and temperature (for details see method section).

In order to demonstrate SI unit tracability and resolution of the system, we investigate pressure broadening of the ammonia molecule by introducing a vacuum tube into the spectroscopy setup, adding a 25% ammonia solution to the tube and evacuating to pressures between 0.018 mbar and 20.6 mbar (for more details on the measurement setup please consider the supplemental information). The absorption line of ammonia is close to 572.5 GHz (Fig. 7c) and has a linewidth of several GHz at atmospheric pressure and room temperature. Upon reducing the pressure it becomes limited by the temperature broadened linewidth of 1.72 MHz for pressures below 0.1 mbar [39] (Fig. 7e, orange curve). The blue curve in Fig. 7c illustrates the measured absorbance for a pressure of 0.041 mbar with a frequency resolution of 24 kHz and averaged over 6 measurements. The black curve is a Voigt fit to the measurement data and the orange line the calculated absorption from the HITRAN

database [39] with a frequency resolution of 100 kHz. The measured peak position is 80 kHz offset to the calculated HITRAN peak position confirming the accuracy and reference capability of the LO. The measured linewidth with 2.4 MHz is well resolved and slightly larger than the theoretical Doppler limited linewidth of 1.72 MHz which may be attributed to foreign broadening caused by water molecules present from the ammonia solution. With increasing pressure in the tube, i.e. at 16 mbar (Fig. 7d), the measured linewidth increases in good agreement with the calculated absorption (Fig. 7e) yet faces an up-shift for the FWHM resulting from the foreign broadening. Fixing the Gaussian component of the Voigt fit, as we do not expect a temperature change, the measured Lorentzian part, responsible for the pressure broadening, (represented by Γ in Fig. 7f) increases similar to the HITRAN-calculated value with a similar upshift as the FWHM.” (lines 284 – 317)

“PRESSURE AND TEMPERATURE BROADENING

The lifetime of the rotational state τ of a molecule limits its natural linewidth (typ. 10-100s kHz) and features a Lorentzian spectral distribution $p(f)$,

$$p(f) = \frac{A}{(f-f_0)^2 + \left(\frac{1}{\pi\tau}\right)}. \quad (7)$$

When two gas molecules collide or their dipole moments perturb each other during a pass by maneuver, the lifetime of the rotational state is lowered. The more molecules per volume are present (i.e. the higher the pressure) the shorter the respective lifetime and the larger the linewidth according to Eq. (7). The pressure broadened linewidth is directly proportional to the pressure, has a Lorentzian shape and reaches linewidths in the few GHz range at normal pressure [49]. We remark that heavier molecules have larger impact such that foreign gases will cause a different linewidth as a pure gas. For the case of ammonia in water shown here, there is only minor alteration as water (18 u) and ammonia (18 u) have very similar molecular mass, though the amplitude, A in Eq. (7), obviously becomes smaller the more water is present. In ambient conditions the absorption also experiences Doppler broadening caused by thermal motion of the gas molecules towards and away from the THz source. The Doppler broadening Δf_D is given by [50]

$$\Delta f_D = f \sqrt{\frac{8 \ln(2) k_B T}{m c^2}}, \quad (8)$$

where f is the center frequency, k_B the Boltzmann constant, T the temperature and m the molecule’s mass. Based on statistical motion of the molecules, Doppler broadening features a Gaussian line shape. The theoretical Doppler limit for ammonia is 1.71 MHz at the 572.5 GHz line at a room temperature of 23 °C.

For the measurements with the ammonia solution under vacuum we used a spectroscopy setup with a vacuum tube in the THz beam. Beamforming equipment in the form of two lenses and photoconductive transmitter and receiver were outside the vacuum tube. A combined membrane pump with a turbomolecular pump enabled pressures as low as 0.001 mbar within the tube. The ammonia solution is added in an extended part of the tube that is separated by a needle valve. For the measurements the needle valve is opened slightly to let ammonia vapor into the tube and depending on the pressure the pumps closed off of the tube. Each measurement consists of three runs of five seconds where each run contains one up and one down scan. During each measurement we monitored the pressure with a pressure gauge. The pressure within the tube changed on average by a factor of 1.5 as the tube is quite leaky. Therefore, each measurement is compared to the average pressure within its measurement time window. The tube does not contain heating or cooling equipment and was left at room temperature (23 °C) limiting the measurements to the Doppler broadened spectra. The absorption coefficient α is calculated by

$$\alpha = -\frac{\ln\left(\frac{I_{NH_3}}{I_0}\right)}{d}, \quad (9)$$

Where I_0 is the intensity of the reference measurement, I_{NH_3} the ammonia measurement and $d = 33$ cm the mean THz path length inside the vacuum tube.” (lines 474-536)

“AMMONIA ABSORPTION SPECTROSCOPY

For the absorption spectroscopy we applied the LO to both the photoconductive transmitter and receiver (Fig. S3). These were positioned opposite each other, each with a TPX-lens for beam forming. In between the lenses we positioned an evacuated cell that can be filled with a gas under test at a desired pressure. The vacuum cell has two fused silica windows positioned under their Brewster angle to reduce reflections. Separated by a needle valve, we attached a probe volume to the vacuum cell where we deposited a drop of ammonia solution. On the other side the vacuum tube connects to a flooding vent, a pressure gauge and a valve to the pump. The pump is a combined unit of a membrane pump and a turbomolecular pump. During the measurements the pressures of the pressure gauge are logged. A bias of -1.2 V is applied to the photodiode transmitter. For the measurements the output of the receiver is amplified by $3.3 \cdot 10^5$ V/A in a TEM Messtechnik PDA-S and acquired by a LeCroy HDO oscilloscope at 500 kSa/s. For the measurement the LO is swept for 15 GHz up and down surrounding the ammonia absorption peak. Each measurement is the average of the Hilbert transformation of 3 up and down sweeps.

Before the start of the measurements we evacuated the cell to pressures below 0.01 mbar and partially the probe chamber to remove remaining air. The finite gas pressure of water and ammonia filled the probe volume with an ammonia-water mixture. Care was taken to slowly flood the gas chamber with the ammonia-water gas mixture by slowly opening the needle valve with the pumps running and connected to the vacuum chamber. Except for the lowest pressures, we cut off the vacuum pumps after approximately 2 minutes and slowly let more probe into the test cell. The leak tightness of the vacuum cell is more in the order of 10 mbar over the course of 30 minutes, i.e. the pressure-broadened linewidths contains some foreign broadening by leakage oxygen and nitrogen into the chamber.” (Supplementals lines 65-86)

Reviewer #2, concern #2: I would better comment the poor signal-to-noise ratios shown in Fig. 2b and 5b. This seems to conflict with the good noise figures shown in Table I.

Author response: Table I shows the noise floor for the best realization of the system, where source and receiver are phase-locked, common noise is almost perfectly cancelled, and all spectral power is essentially collected in a single spectral bin. Instead, Fig. 2b shows the case of an unlocked CW system with MHz bandwidth whose spectrum is analyzed by the comb-referenced system. For spectral analysis, the source under test and the characterization system must not be phase-locked in order to resolve the true spectrum. The CW source's power is spread out over its whole linewidth of up to 50 MHz, resulting in about 7.3 orders of magnitude lower signal-to-noise ratio (SNR) compared to if all spectral power was collected in a single 2.3 Hz frequency bin as in Fig. 6b).

Fig. 5 b shows a partially locked case, where the same GPS reference is used but two different OXCOs. As seen from the orange line in Fig. 5c, still quite some non-common noise remains between the source and the receiver widening the linewidth and reducing the SNR. Only in Fig. 6 we show a result with almost perfect common noise cancellation that enabled increasing the spectral coverage from 4.75 THz (Fig. 5b shows 3.25 THz) to 6.5 THz (Fig. 6b). Please also consult our response to reviewer #3 concern #3 for similar changes.

Author action: We added some more detail within the text describing the figures (l. 162-168 and 243-250) in order to clarify the differences and added a section on the SNR into the methods.

"The peaks at both frequencies are well resolved with a peak power of -62 dBm and a noise power below -87 dBm at 532 GHz and a peak power of -75.5 dBm and a noise power of -78.5 dBm at 1,782 GHz. The increase in noise floor is the result of reduced responsivity within the photoconductive receiver in the PSA due to RC and lifetime roll-off effects [12, 27]. In the PD transmitter similar roll-off effects are present leading to the reduction in signal power [28]. At 1,782 GHz the larger linewidth spreads the power of the THz source out more, further reducing the spectral power density recorded at each frequency point." (lines 162-168)

"We remark that the noise floor at 3.25 THz in Fig. 5 b) of -107.8 dB (yet at an equivalent noise bandwidth of 1 Hz) shows the noise floor in direct vicinity to the signal, the actual noise floor at an offset frequency of 35 kHz to the signal is at -112 dB, i.e. another 4.2 dB lower. At a frequency resolution of 1 Hz this value is slightly larger than the estimated DANL of -117.2 dBm/Hz in Fig. 4. This is due to a combination of 1) the noise floor of the post detection electronics that play an increased role with the Fourier transformation mode used here and the frequency resolution of 1 Hz, 2) ground loop effects that influence the noise floor in our laboratory setup and 3) increasing absorbance and reflectance from the InP substrate [33]." (lines 243-250)

"SIGNAL-TO-NOISE RATIO

The signal-to-noise ratio (SNR) within the measurements is dependent on the noise floor and responsivity of the receiver side, the THz power from the emitter and the linewidth of the signal-under-test and of the LO. All of the sources shown in this manuscript use photomixers (pin diodes or photoconductors) to generate the THz signal. Independent of the used optical signal (pulsed or CW), the power of the signal decreases with increasing frequency for frequencies higher than the roll-off 3dB-frequencies. For the pulsed variant, additionally, fewer optical modes mix with each other with increasing frequency, further reducing the signal power. The signal path consists in most cases of two parabolic mirrors in a U-configuration, Si lenses and InP substrate within the photomixer's package and a free-space path of less than 0.5 m. Along the signal path losses, reflection and absorption (i.e. water vapor, see also Fig. 7b) may occur reducing the signal strength inbound to the receiver. The photocurrent generated by the receiver depends on the strength of the incoming electrical field (see

Eq. (2)) and the LO frequency. Roll-off effects within the photomixer reduce its responsivity with increasing LO frequency. The noise current of the photomixer is mostly independent of the LO frequency. The lower responsivities at higher frequencies require thus larger signals for successful detection. Additionally, the noise floor is directly proportional to the measurement bandwidth. Smaller measurement bandwidths (i.e. longer integration times) allow for the detection of smaller signals. The system specific noise floor is therefore frequently given in the unit dB/Hz.

The bandwidth is limited on the lower end by the LO linewidth. The system essentially measures the spectral density, i.e. the power of the signal within a measurement bandwidth (frequency bin). Spectrally wide LOs or signals -such as the one in Fig. 2- are distributed over many frequency bins, reducing the power per bin and thus the signal to noise ratio. As the responsivity of the photoconductors drop with increasing frequency smaller SNRs lead to smaller usable bandwidths, ranging from DC to the maximum still detectable signal. This lead to the result that the ~50 MHz wide CW signal featured the lowest usable bandwidth, while the usable bandwidth increased the better the relative stability of SUT and LO, yet with 6.5 THz bandwidth for close to perfect relative stability where all spectral power is concentrated within a 2 Hz frequency bin. “ (lines 474 – 500)

Reviewer #2, concern #3: There is an intensive advertisement in Figs. 1, 3, 5, 6. of the frequency comb source produced by Toptica, namely the DFC core: name and model of the used comb source should definitely appear just once in the text (and not in the figures).

Author response: We are sorry, we just wanted to clarify the system composition in order to make our findings reproducible.

Author action: We have replaced the wording DFC core with frequency comb (FC) in basically all cases.

Reviewer #3, concern #1: I find very interesting the concept of referencing the LO to an Oven Controlled Crystal (Xtal) Oscillator (OCXO), which is itself referenced to a 10 MHz reference signal locked to GPS. This discussion and the implication of this scheme are a bit hidden in the current version of the manuscript. I think they could be emphasized to amplify the impact of this work.

Author response: Both CW lasers are locked to the same pulsed laser whose phase noise the CW laser inherit. The phase stability between the different modes of a pulsed laser are excellent even in an unreferenced pulsed laser. Locking the repetition rate – or a higher order of it, like done here – to a low-noise radio frequency (RF) signal further reduces the relative noise between the two CW signals. At the same time, the difference frequency becomes relatable to the RF reference. Referencing the OCXO to a 10 MHz GPS reference signal should give the repetition rate of the pulsed laser and further the LO frequency the same Allan deviation as the GPS reference signal. A 10 MHz reference is a common choice for test and measurement equipment. This gives an absolute frequency accuracy of approximately Hz level even at multiple THz. For further details on this topic please view ref. <https://doi.org/10.1063/5.0217898> and ref. 29. The direct referencing to GPS allows for metrological applications which we added to the manuscript. Please also see reviewer #2, concern #1.

Author action: We updated the information on the locking mechanism and added further references. To showcase the absolute frequency accuracy stemming from the referencing of the LO to the GPS signal, we investigated the absorption line of an ammonia solution at its absorption line close to 572.5 GHz. We achieved excellent agreement with literature data. The details are part of the homodyne CW-THz-spectroscopy section including Fig. 7.

“Within all measurements shown in this manuscript the LO is additionally referenced via an oven-controlled crystal oscillator (OCXO) to a global positioning system (GPS) signal. The OCXO generates an 800 MHz reference signal that phase-locks to the 4th harmonic of the repetition rate of the FC. The additional referencing of the OCXO to the GPS gives the LO excellent absolute frequency accuracy which we verify in a later section.” (lines 131-135)

“GPS referencing also allows for comparable measurements of the same signal at different locations and times.” (lines 228-230)

“In order to demonstrate SI unit tracability and resolution of the system, we investigate pressure broadening of the ammonia molecule by introducing a vacuum tube into the spectroscopy setup, adding a 25% ammonia solution to the tube and evacuating to pressures between 0.018 mbar and 20.6 mbar (for more details on the measurement setup please consider the supplemental information). The absorption line of ammonia is close to 572.5 GHz (Fig. 7c) and has a linewidth of several GHz at atmospheric pressure and room temperature. Upon reducing the pressure it becomes limited by the temperature broadened linewidth of 1.72 MHz for pressures below 0.1 mbar [39] (Fig. 7e, orange curve). The blue curve in Fig. 7c illustrates the measured absorbance for a pressure of 0.041 mbar with a frequency resolution of 24 kHz and averaged over 6 measurements. The black curve is a Voigt fit to the measurement data and the orange line the calculated absorption from the HITRAN database [39] with a frequency resolution of 100 kHz. The measured peak position is 80 kHz offset to the calculated HITRAN peak position confirming the accuracy and reference capability of the LO. The measured linewidth with 2.4 MHz is well resolved and slightly larger than the theoretical Doppler limited linewidth of 1.72 MHz which may be attributed to foreign broadening caused by water molecules present from the ammonia solution. With increasing pressure in the tube, i.e. at 16 mbar (Fig. 7d), the measured linewidth increases in good agreement with the calculated absorption (Fig. 7e) yet faces an up-shift for the FWHM resulting from the foreign broadening. Fixing the Gaussian component of the Voigt fit, as we do not expect a temperature change, the measured Lorentzian part,

responsible for the pressure broadening, (represented by Γ in Fig. 7f) increases similar to the HITRAN-calculated value with a similar upshift as the FWHM.” (lines 298-317)

“The referencing happens with a control circuit that matches the 4th harmonic of the repetition rate of the FC (200 MHz) to the frequency of the OCXO (800 MHz).” (lines 392-393)

Reviewer #3, concern #2: The authors claim that their receiver is capable of resolving frequencies up to 10 THz. It's not clear if this is possible though, since it has not been demonstrated in this work. Carrier lifetime in the photoconductor material could notably impose a fundamental spectral limit. The authors also mention that "the Reststrahlenband of the InP substrate of the photoconductive mixer will limit the frequency coverage to about 7 THz". This might also be a fundamental limit as it is not clear how the receiver would perform without this substrate. Therefore, the authors must refer to a maximum frequency detection reaching 6.5 THz (which is demonstrated in this paper) instead of 10 THz. The authors must also change the wording in their abstract which also seems to indicate that their receiver operates up to 10 THz: "We present an ultra-wideband, continuous-wave photonic receiver with kHz-level spectral resolution in the terahertz domain (100 GHz-10 THz)".

Author response: The mentioned 10 THz relates to the definition of the terahertz domain and does not show the performance of our measurements. On a side note we would like to mention that some of the authors have demonstrated photoconductor performance beyond 10 THz with devices where the substrate (and hence absorption by its Reststrahlenband) was essentially removed [39]. This would prove that the photoconductor's lifetime performance would suffice to cover the whole THz spectral range.

Author action: We removed the number "10 THz" in the abstract to prevent confusion.

Reviewer #3, concern #3: The paper mentions some limitations of their receiver due to the spectral fluence of the source, e.g. at Line 149. A more in-depth discussion would be necessary to determine how the fluence of the source may limit the performances of their receiver.

Author response: The signal of the source may be described with its total power and its spectral linewidth. Signals with a larger linewidth spread their power over a larger spectrum than signals with a small linewidth. The peak spectral power density of the signal with small linewidth will be correspondingly higher. Both the photomixer used for the transmitters and the receiver have roll-off effects (RC and lifetime) that deteriorate their performance towards higher frequencies. Please also view our response to reviewer #2 concern #2 for similar changes.

Author action: We added a section on signal-to-noise ratio to the method section.

“SIGNAL-TO-NOISE RATIO

The signal-to-noise ratio (SNR) within the measurements is dependent on the noise floor and responsivity of the receiver side, the THz power from the emitter and the linewidth of the signal-under-test and of the LO. All of the sources shown in this manuscript use photomixers (pin diodes or photoconductors) to generate the THz signal. Independent of the used optical signal (pulsed or CW), the power of the signal decreases with increasing frequency for frequencies higher than the roll-off 3dB-frequencies. For the pulsed variant, additionally, fewer optical modes mix with each other with increasing frequency, further reducing the signal power. The signal path consists in most cases of two parabolic mirrors in a U-configuration, Si lenses and InP substrate within the photomixer's package and a free-space path of less than 0.5 m. Along the signal path losses, reflection and absorption (i.e. water vapor, see also Fig. 7b) may occur reducing the signal strength inbound to the receiver. The photocurrent generated by the receiver depends on the strength of the incoming electrical field (see Eq. (2)) and the LO frequency. Roll-off effects within the photomixer reduce its responsivity with increasing LO frequency. The noise current of the photomixer is mostly independent of the LO frequency. The lower responsivities at higher frequencies require thus larger signals for successful detection. Additionally, the noise floor is directly proportional to the measurement bandwidth. Smaller measurement bandwidths (i.e. longer integration times) allow for the detection of smaller signals. The system specific noise floor is therefore frequently given in the unit dB/Hz.

The bandwidth is limited on the lower end by the LO linewidth. The system essentially measures the spectral density, i.e. the power of the signal within a measurement bandwidth (frequency bin). Spectrally wide LOs or signals -such as the one in Fig. 2- are distributed over many frequency bins, reducing the power per bin and thus the signal to noise ratio. As the responsivity of the photoconductors drop with increasing frequency smaller SNRs lead to smaller usable bandwidths, ranging from DC to the maximum still detectable signal. This lead to the result that the ~50 MHz wide CW signal featured the lowest usable bandwidth, while the usable bandwidth increased the better the relative stability of SUT and LO, yet with 6.5 THz bandwidth for close to perfect relative stability where all spectral power is concentrated within a 2 Hz frequency bin.” (lines 474-500)

Reviewer #3, concern #4: In the section “Spectrum analyzer for unreferenced sources”, the authors should show the full spectrum from 0.1 to 6.5 THz in addition to the resolution of a single comb line in Fig. 3b.

Author response: The power of the investigated unreferenced source has a linewidth of around 50 MHz. Its THz power is spread over the linewidth. Just looking at the linewidth the spectral power density of the source will be more than 7 orders of magnitude lower compared to the locked case in Fig. 6. A CW source ideally only emits at one frequency. Measuring a full spectrum from 100 GHz to 6.5 THz will contain noise for almost all of the spectrum except the 50 MHz where the system emits. Graphically, the peak will be hard to find and the shape of the line will not be visible to the reader considering the fact that such a measurement would contain at least 260 000 data points if a maximum step size of 25 MHz (Shannon Hartley limit for a 50 MHz linewidth) is chosen. We therefore opted not to measure the whole spectrum. Depending on the chosen resolution bandwidth, long measurements will be limited by the number of data points the acquisition card can measure. Simultaneously, Fig. 7 shows measurements with a total measurement bandwidth of more than 3 THz to emphasize the capabilities of the LO.

Author action: The section “principle of operation” contains information on the tunability of the LO.

“By simply sweeping the LO frequency, we can record the spectrum of any unknown source under test at a resolution defined by the excellent spectral purity of the FC-referenced LO. Given that the source’s power is high enough to be detected, the maximum tuning range is limited only by the tuning range of the LO (>10 THz) [29] and the sensitivity of the photoconductor which becomes smaller the higher the frequency. The frequency coverage of the used photoconductors is limited by the Reststrahlenband of the InP substrate to 7 THz. Removing the substrate allows for coverage beyond 10 THz [33].”

Reviewer #3, concern #5: The authors have showed homodyne CW THz-spectroscopy in the paper with an integration time of 50 μ s where the signal is distinguishable until 3.5 THz. One of their previous articles show a spectrum reaching 5 THz with an integration time of 300 ms (Ref [26]). I recommend using 300 ms here also to allow for a better comparison between this configuration and the previous one used in Ref. 26.

Author response: We thank the reviewer for the suggestion. The measurement time increases by a factor of 60,000 if using 300 ms integration time and if done with the same spacing resulting in a total measurement over multiple days. As we had to bring equipment from three labs together that was also needed for further experiments, we simply could not take this much time for a single measurement. The integration time and its effect on the bandwidth of photomixing systems has been described in plenty detail in the literature (see [34] and <https://doi.org/10.1063/5.0217898>). The long measurement would not have brought new fundamental information.

Author action: We added additional literature to the manuscript for a better understanding of our spectroscopy results.

“Longer averaging allows for accessing even higher frequencies [27, 34].” (lines 292-293)

Reviewer #3, concern #6: Please check: the abbreviation of oven-controlled crystal (xtal) oscillator should be OCXO, not OXCO.

Author response: We thank the reviewer for noticing the incorrect abbreviation.

Author action: Replaced all instances of OXCO with OCXO.

Reviewer #3, concern #7: Regarding the sentence: “While a spectrum analyzer strictly requires that the LO and the signal under test are independent in order to measure the true signal shape, noise and stability, this independence is undesired in spectrometers.”, the authors should provide more information to explain why they are making these assumptions. It is advantageous (but maybe not necessary) that a spectrum analyzer is independent from the signal it analyzes, but it is less clear why this independence is “undesired” for spectrometers. Most spectrometers don’t have any “dependence” with the source they characterize.

Author response: Generally speaking both the spectrum analyzer and the spectrometer may be used with or without referencing the signal under test to the LO. If a spectrum analyzer does not use referencing, the SUT will be correctly represented as no common phase noise exists, assuming the phase noise of the SUT dominates. If the source to be analyzed is spectrally locked to the analyzer instead, most of the phase noise is common-mode and thus suppressed. The recorded linewidth and drifts, e.g., will then always be smaller than the true linewidth and drift, which is obvious for drifts: When source and receiver drift synchronously, which they would if phase-locked, then drifts/frequency stability cannot be analyzed at all.

The reviewer is completely right that phase-locking is no stringent requirement for spectrometers and even not possible if direct detectors rather than mixers are used. Phase-locking can, however, dramatically reduce common noise as demonstrated in this manuscript for homodyne or heterodyne spectrometers: The spectrometer variant discussed in that section benefits from locking as the receiver measures primarily at the frequency of the LO.

Author action: We improved the paragraphs of our spectrometer and spectroscopy variant to better showcase the benefit of the shared LO.

“In order to measure the true signal shape, noise and stability, a spectrum analyzer strictly requires that the LO and the signal under test are independent. In contrast, spectrometers used for investigating devices under test benefit from locking the LO with the signal under test. Channel sounding with locking to the same GPS signal already reduces the phase noise but for even better noise suppression both source and receiver can be locked to the same OCXO (Fig. 6a). Fig. 6b) shows the spectrum of the pulsed source at a frequency of 6.5 THz with a combined (relative) linewidth of only 2.3 ± 1.2 Hz and a signal-to-noise ratio of at least 7 dB for 100 averages. Fig. 5c) (blue line) also shows that the combined linewidth only slightly depends on the THz frequency and remains below 3 Hz over the whole frequency range.” (lines 261-269)

“Homodyne spectrometers profit from using the same LO signal for both the transmitter and the receiver, reducing common noise and enabling phase measurements.” (lines 285-286)

Reviewer #3, concern #8: This sentence: “Fig. 6b) shows that locking to a common reference allows to extend to a maximum frequency of 6.5 THz”, it’s not clear. First, Fig. 6b only shows a power signal, not a comparison for locked and unlocked signals. Also, I believe that the authors are making a confusing shortcut: using the same OCXO probably leads to an improved signal-to-noise, this in turns might lead to an increase in the detection spectral range. This should be clarified.

Author response: The reviewer is correct that using the same OCXO leads to an improved SNR. Locking to a common reference increases the common noise between the optical signals which in turn reduces the relative phase noise between both signals. In this case the signal is concentrated in a smaller linewidth boosting the signal-to-noise ratio. This is easy to see by comparing the linewidth in Fig. 6b of 2.3 Hz (common-noise and drifts rejected) to the free-running linewidth we extrapolated for unlocked sources to be around 800 Hz (c.f. line 272). This enables the measurement of higher frequencies as all photomixing systems show a decay in SNR towards higher frequency. If one starts with a higher SNR in the first place because of common noise rejection, then the system hits the noise floor at a higher frequency.

Author action: We updated lines 270 to 277 for clarity.

“The high correlation of both lasers drastically improves the signal to noise ratio, highly beneficial for spectroscopic applications. We estimate the power in the mode at 6.5 THz to 7 pW which is now concentrated in a 2 Hz frequency bin as opposed to an 800 Hz linewidth in the non-referenced configuration, improving the signal to noise ratio by about a factor of 400. This improvement enabled the extension of the operation range very close to the current frequency limit of the photoconductive mixer, which originates from strong reflection and absorption of the Reststrahlenband of its Indium Phosphide substrate.” (lines 270-277)

Reviewer #3, concern #9: The sentence “Key applications are future high-speed wireless communication, with several THz bands already defined in 6G [1]” could be misleading because the Ref [1] mentions that frequencies between 1 THz and 3 THz have not yet been defined as this spectral region can be used by both active and passive services. Plus, nothing is mentioned in Ref [1] about frequencies above 3 THz.

Author response: Thank you for the note, this was indeed confusing.

Author action: We updated the sentence in line 34.

“Key applications are future high-speed wireless communication, with several bands already defined in 6G in the lower THz region [1] and more to come in follow-up standards, non-destructive testing and quality control [2], as well as spectroscopy [3] and imaging [4], with particular focus on medically or environmentally relevant topics.” (lines 33-36)

Reviewer #3, concern #10: This sentence: “For the highly important field of THz wireless communication an opto-electronic device is even a prerequisite to transform the optical signal, transmitted through a fiber, to a wireless THz signal.” should be removed since the statement is not always true. For example, a nonlinear medium could be used to transform the optical signal into a wireless THz signal.

Author response: Indeed, there may be other options. To the knowledge of the authors, however, opto-electronic devices are the by far best technological solution so far for non-pulsed signals. Non-linear media do indeed work also at CW but are very inefficient, particularly at frequencies of interest to 6G that are far away from the optical tone and thus heavily impacted by the Manley-Rowe limit. The only continuous-wave case the authors are aware of where a non-linear medium could result in excellent up-conversion or down-conversion is when placing the medium in a resonator, ideally a double-resonant (THz, NIR) setup.

Author action: We changed lines 49 to 57 to account for options except the opto-electronic devices and added reference [10].

“Nonlinear media and photomixers bridge this gap by combining electronic with photonic approaches. Nonlinear media benefit from high THz powers and large bandwidths, yet lack in efficiency. Resonators increase the coupling efficiency at the expense of tunability [10]. Photomixers, i.e. photodiodes (PDs) or photoconductors (see section photonic mixer in methods), use two laser signals for THz generation and detection [11, 12]. Meanwhile, photonic approaches transition into the electronic domains, e.g. in the form of a low phase noise photonic clock signal [13], or photonic local oscillators [14], and mixed electronic-photonic integrated circuits (EPICs) [15]. The highly important field of wireless THz communication benefits from opto-electronic devices directly transforming optical signals, transmitted through a fiber, to a wireless THz signal.” (lines 49-57)

Reviewer #3, concern #11: This sentence could be misleading: “The extenders double or triple the microwave baseband frequency several times up to the desired THz frequency or from the THz frequency down to the base band [15].” Ref 15 demonstrates this concept up to 400 GHz and intrinsic carrier response time in materials still limit these applications to about 1 THz only.

Author response: The reviewer is correct that the frequency multiplication from electronic sources is limited. Commercially available extender modules reach 1.5 THz.

Author action: We adjusted the sentence before in line 59 to state a limit on its current commercial availability.

“They consist of a microwave backbone system equipped with frequency extenders, currently available up to 1.5 THz.” (line 59-60)

Reviewer #3, concern #12: There's no verb in this sentence: "For example, a Hilbert-transform spectrum analyzer based on a high-T_c Josephson junction [17] with a high scanning speed of up to 4 67 THz/s and a frequency coverage of at least 1 THz, yet only reaching a resolution of 50 MHz."

Author response: We thank the author for noticing the missing verb and apologize for it.

Author action: We changed the sentence to include a verb.

"For example, a Hilbert-transform spectrum analyzer based on a high-T_c Josephson junction [18] achieves a high scanning speed of up to 4 THz/s and a frequency coverage of at least 1 THz, yet only a resolution of 50 MHz." (lines 70-72)

Reviewer #3, concern #13: This sentence is long and confusing: “These utilize continuous-wave (CW) lasers as local oscillator (LO) – namely, an electro-optical comb based on an electro-optical phase modulator at telecom wavelengths with a frequency coverage of up to 1 THz and a resolution of 1 Hz [22], free-running distributed feedback (DFB) laser diodes with a difference frequency coverage of up to 4.5 THz at a wavelength of 780 nm [25] and a resolution of 1.2 MHz at a center wavelength around 1550 nm [23] or DFB laser diodes at 850 nm referenced to an electronic THz signal with a frequency coverage of 1 THz and a resolution of 100 Hz [24].”

Author response: We are sorry for implementing a long and confusing sentence.

Author action: We separated the sentence (lines 82-87).

“These utilize continuous-wave (CW) lasers as a local oscillator (LO). An electro-optical comb based on an electro-optical phase modulator at telecom wavelengths covers a frequency range of up to 1 THz with a resolution 1 Hz [23]. Free-running distributed feedback (DFB) laser diodes enable a difference frequency coverage of up to 4.5 THz at a wavelength of 780 nm [26], a resolution of 1.2 MHz at a center wavelength around 1,550 nm [24]. DFB laser diodes at 850 nm referenced to an electronic THz signal reduce the resolution to 100 Hz and the frequency coverage of 1 THz [25].” (lines 82-87)

Reviewer #4:

Author response: We thank you for carefully reviewing the manuscript and welcome the process of co-reviewing the manuscript and hope that it benefits you in your research career.

Response to reviewers on Nature Communications manuscript NCOMMS-24-72395-A

Dear Reviewers,

We sincerely thank you for thoroughly reviewing the manuscript and the comments and suggestions. Please find below a detailed response to each of the remarks individually. We excuse for the long duration of the revision. We hope to have improved the manuscript to allow for further consideration.

Yours sincerely,

Benedikt Krause

(Corresponding author, in the name of all authors)

Reviewer #1, Concern #1: The authors' work is very much related to the paper "Photonic comb-rooted synthesis of ultra-stable terahertz frequencies" published by DC Shin, BS Kim, H Jang, YJ Kim, SW Kim in Nature Communications (2023)14:790. Nonetheless, the 2023 Nature Comms paper was neither cited nor discussed even in the authors' revised manuscript – It is strongly suggested that the authors describe how their work is differentiated and valued comparatively in terms of the THz generation system setup, frequency instability, phase noise and tuneable range before making final claims on their achieved performance.

Author response: We thank the reviewer for providing a reference on a similar topic. In the reference the frequency comb is locked to an optical cavity and two DFB laser diodes are injection locked to the frequency comb. Additional to the locking, the fiber thermal noise is compensated. This allows a THz linewidth of 2 mHz or less at THz frequencies of 0.1, 0.66 and 1.06 THz. Without the fiber thermal compensation the linewidth broadens to 1.46 Hz. Due to the chosen locking variant, the DFB laser diodes may only lock to the comb lines and therefore only generate THz frequencies spaced apart by multiples of the repetition rate of the frequency comb.

The LO described in this manuscript phase-locks two external cavity laser diodes to a frequency comb. In order to cover any difference frequency up to 10 THz, one of the ECDLs is phase-locked to one of the modes of the frequency comb while the second is phase-locked to a frequency shifted variant of the frequency comb. Tuning the ECDL and the frequency shift of the frequency comb simultaneously achieves continuous tuning with tuning speeds up to 1 THz/s. Sweeping the LO at a rate of 11.8 GHz/s we achieved an accuracy of 120 kHz at a center frequency of 556.936 GHz, with respect to HITRAN data. While keeping the LO fixed we achieved an accuracy and stability of 2 Hz at a center frequency of 6.5 THz in comparison with a harmonic of the repetition rate of a stabilized comb.

Author action: We added different LO / THz frequency synthesizer variants mentioned in the literature to the introduction and a short comparison to the discussion.

"The ambiguity of pulsed comb-based systems can also be overcome by locking continuous-wave lasers to specific comb lines while tuning the comb's repetition rate, yet with a demonstrated resolution of 200 kHz and a tuning range up to 1.28 THz [23]. Likewise, high spectral purity can be achieved by locking the CW lasers to different resonances of a high quality factor cavity [24], with a sub-Hz frequency stability demonstrated at 1.965 GHz and an accuracy of 6 kHz at 556.936 GHz. Coarse tunability of up to 1.2 THz is achieved by hopping between cavity resonances. The gaps between the resonance are closed by fine-tuning the difference frequency with a Mach-Zehnder modulator. Ref. [25] references the CW lasers to a frequency comb which itself is locked to an optical cavity achieving a terahertz frequency stability of 1.46 Hz. Removing the fiber thermal noise improves the stability further to 2 mHz. Due to the referenced frequency comb this setup is limited to THz frequencies spaced apart by the repetition rate of the FC with a demonstrated frequency coverage of 1.1 THz. So far, these high purity continuous-wave systems have not been demonstrated as spectrum analyzers and their maximum demonstrated operation frequency is, to date, below those of commercial ESAs (1.5 THz)." (lines 82-94)

"The additional referencing of the OXCO to the GPS gives the LO absolute frequency accuracy which we verify in a later section." (lines 147-149)

"The FC-referenced photonic LO performs very similar in frequency stability and accuracy to other FC or cavity referenced photonic THz synthesizers [23, 24]. Accounting for fiber thermal noise could further improve the performance [25]. Yet the system presented in this work combines the stability with by far the fastest tunability option of comparable synthesizers." (lines 355-359)

“Therefore, the systems give access to low pressure trace gas detection of Doppler-limited lines over a large fraction of the THz range which we showed exemplary by investigating the water vapor absorption line at 556.936 GHz.” (lines 373-375)

- [23] Mouret, G. *et al.* “THz photomixing synthesizer based on a fiber frequency comb”, *Optics express* **17**, 24, 22031-22040 (2009).
- [24] Djevahirdjian, L. *et al.* “Frequency stable and low phase noise THz synthesis for precision spectroscopy”, *Nat. Commun.* **14**, 7162 (2023).
- [25] Shin, D.-C., Kim, B.S., Jang, H. Kim, Y.-J., Kim, S.-W. “Photonic comb-rooted synthesis of ultra-stable terahertz frequencies”, *Nat. Commun.* **14**, 790 (2023).

Reviewer #2: The authors have thoroughly addressed all reviewers' concerns and well complemented the paper with missing measurements, clarifications, extensions. I am in favor of the publication of the manuscript in Nat Comm.

Author response: We thank the reviewer for proofreading our manuscript and for the helpful comments in the previous revision.

Reviewer #3, Concern #1: I am still not convinced, based on experimental evidence, that the performance beyond 10 THz is achievable. This was my “concern #2”. The authors answered: “[...] some of the authors have demonstrated photoconductor performance beyond 10 THz with devices where the substrate (and hence absorption by its Reststrahlenband) was essentially removed [39]”. I couldn’t find this photoconductor performance demonstration beyond 10 THz in reference [39], which is Gordon et al. J. Quant. Spectrosc. Radiat. Transfer 277, 107949 (2022). The authors also write in the text: “Removing the substrate allows for coverage beyond 10 THz [33].” But reference [33], which is OE 20, 23896 (2022), doesn’t show (or even claim) to experimentally demonstrate THz detection beyond 10 THz. In that previous work, the measured signals are very close to the noise level after 9.5 THz. Because of the results presented in this paper and in previous work referenced by the authors [33, 39], I recommend removing any claims on the performance of these devices exceeding 10 THz when the substrate is removed.

Author response: Firstly, we apologize for the confusion in the references. We shifted the reference for OE 20, 23896 (2022) late into the response and forgot to correct it in the response letter. Secondly, the reviewer is correct that mentioned reference experimentally only shows frequencies up to 10 THz. For better understanding, we changed the source to Proc. SPIE 12885, Terahertz, RF, Millimeter, and Submillimeter Wave Technology and Applications XVII, 1288503 (2024) as this one uses newer data despite being in a less recognized journal. Fig. 3b (see below) of the reference shows a frequency coverage up to 10 THz with a remaining dynamic range of at least 10 dB at 10 THz. Therefore showcasing that beyond 10 THz should be possible.

Fig. 3b) from Proc. SPIE 12885, Terahertz, RF, Millimeter, and Submillimeter Wave Technology and Applications XVII, 1288503 (2024): “...(b) Fourier transformed pulse spectrum and measured noise level: A peak dynamic range of 105 dB and a bandwidth of 10 THz are achieved within 60 s acquisition time (1000 averages).”

Author action: We corrected and double checked all references to make sure they are in the correct position. We also adapted the sections to state that a frequency coverage of at least 10 THz is possible with the system by replacing the photoconductor substrates.

“The frequency coverage of the used photoconductors is limited by the Reststrahlenband of the InP substrate to 7 THz. Removing the substrate allows for coverage of at least 10 THz [36].” (lines 142-144)

“By removing the substrate the frequency coverage of the photoconductive mixer can be extended, as demonstrated in ref. [36], to cover at least 10 THz, equaling the full tuning range of the FC-referenced photonic LO.” (lines 353-355)

Reviewer #3, Concern #2: The authors propose a plausible cause to explain the slightly larger spectral linewidth of ammonia, which they attribute to the presence of contaminating water molecules. However, the THz spectrum should allow them to test directly this hypothesis by monitoring spectral components at water molecule absorption peaks. The authors should employ this technique to confirm their hypothesis or provide a justified explanation for any limitations preventing its implementation.

Author response: Indeed, we had underestimated the number and influence of several error sources on the ammonia spectra. We added a droplet of ammonia water and assumed to essentially see the native ammonia line with some foreign broadening mostly by air. Unfortunately, we cannot tell how much water vapor (from the evaporating droplet) and how much rest air were in the gas phase in the vacuum chamber. So we cannot tell for ammonia the exact foreign broadening. We did not just want to generate a fitted result by having all other concentrations as fit parameters as this proves essentially nothing (too many fit parameters, once can eventually get everything somehow in agreement). Further known error sources of this measurement are: a) the pressure gauge that was not calibrated before the measurements. We attribute an error of 20 % to the pressure measurements, b) Also we have no idea about reactions with the chamber wall resulting in difficulties to predict the partial pressures of the individual gas components, c) The concentrations in the gas phase of ammonia and water will not be identical to the liquid phase, d) Last but not least, the leakage of laboratory air into the chamber may increase the amount of water vapor.

As we just intended to provide a proof for the excellent resolution of the system, we followed the advice of the reviewer and inspected a water line instead. Due to the close proximity between the ammonia absorption line (572.5 GHz) and the water line (557 GHz), we had indeed captured both within the same measurements. The measurement of the water line at low pressures shows much less errors for the following reasons: Within our setup the transmitter and the receiver are mounted outside the pressure chamber. Any humidity remaining in the surrounding air will influence the linewidth of the signal. For this reason the area surrounding the transmitter and receiver were flooded with dry air essentially removing any water vapor in the free space THz path outside the pressure chamber, providing just a very minor error source. The amount of ammonia in the gas phase is (expected to be) much smaller than water vapor for a 25% ammonia water. As water and ammonia have almost identical molar mass eigen broadening and foreign broadening are very similar. Essentially, we measure water vapor foreign broadened by air. For the calculations we assumed a water vapor content of 25 % with an error of up to 10 %. At the lowest investigated pressures, errors in the pressure gauging have less influence. We thus get much better agreement with HITRAN data for the water line as the new Fig. 7c) shows for a pressure of 0.032 mbar.

Fig. 7: c) Absorption measurement (blue curve) and Voigt fit (black curve) with simulated absorption (orange curve) of water vapor at 0.032 mbar. d) Measured (blue circles) and simulated (orange triangles) absorption full width half maximum (FWHM).

The measured water vapor absorption spectra (blue line) with a Voigt fit (black line) and the calculated data from HITRAN (orange line) for the same pressure agree very well. The fitted center frequency is 556.936 12 GHz and the calculated one is 556.936 GHz showing a difference of 120 kHz. We purposefully only show the water vapor results for the lowest pressures where the lines are mostly Doppler broadened and the uncertainty of the pressure error has less influence. In Fig. 7d we compare with measured FWHM to the HITRAN FWHM. For the pressures between 0.020 mbar and 0.046 mbar the difference in FWHM are less than 45 kHz between the calculated and the measured spectra. For both peak position and FWHM, the deviation with respect to the HITRAN data is just a fraction of the FWHM, namely a relative deviation of 215 ppb for the peak position and only 1.1% for the FWHM at 0.032 mbar.

Author action: We exchanged the ammonia data shown within the previous version of the manuscript with the water vapor data for the pressure ranges with a low measurement uncertainty.

“In order to demonstrate the accuracy and resolution of the system, we investigate an almost Doppler-limited line of the water molecule by introducing a vacuum tube into the spectroscopy setup, adding a drop of ammonia water to the tube and evacuating to pressures between 0.020 mbar and 0.046 mbar (for more details on the measurement setup please consider the supplemental information). The absorption line of water vapor is close to 556.936 GHz (Fig. 7c) and has a linewidth of several GHz at atmospheric pressure and room temperature. Upon reducing the pressure it becomes limited by the temperature broadened linewidth of 1.62 MHz [42] (see Eq. (8) in method section) caused by the optical Doppler effect of gas molecules at thermal velocity. The blue curve in Fig. 7c illustrates the measured water vapor absorption for a pressure of 0.032 mbar with a frequency resolution of 24 kHz, a measurement speed of 11.8 GHz/s and averaged over 6 measurements. The black curve is a Voigt fit to the measurement data and the orange line the calculated absorption from the HITRAN database [42] with a frequency resolution of 100 kHz assuming a water vapor to air mixture of $25\% \pm 10\%$. The measured peak position is 120 kHz offset to the calculated peak position, corresponding to a relative deviation with respect to the HITRAN data of 215 ppb. The measured linewidth with 1.81 MHz is well resolved and only 1.1% smaller than the linewidth of 1.83 MHz (Fig. 7d) calculated from HITRAN. Given an estimated error of 20% on the pressure measurements, these data confirm the accuracy and reference capability of the LO.” (lines 312 – 328)

“Therefore, the systems give access to low pressure trace gas detection of Doppler-limited lines over a large fraction of the THz range which we showed exemplary by investigating the water vapor absorption line at 556.936 GHz.” (lines 373-375)

“We remark that heavier molecules have larger impact such that foreign gases will cause a different linewidth as a pure gas. In ambient conditions the absorption also experiences Doppler broadening caused by thermal motion of the gas molecules towards and away from the THz source.” (lines 526-529)

“The theoretical Doppler limit for water vapor is 1.62 MHz at the 556.936 GHz line at a room temperature of 23 °C.

For the measurements with water vapor under vacuum we used a spectroscopy setup with a vacuum tube in the THz beam. Beamforming equipment in the form of two lenses and photoconductive transmitter and receiver were outside the vacuum tube, purged with dry air to prevent water absorption at normal pressure outside the tube. A combined membrane pump with a turbomolecular pump enabled pressures as low as 0.001 mbar within the tube. Additional water in form of an ammonia-water solution is added in an extended part of the tube that is separated by a needle valve. For the measurements the needle valve is opened slightly to let water vapor into the tube and depending on the pressure the pumps closed off of the tube. Each measurement consists of three runs

of five seconds where each run contains one up and one down scan. During each measurement we monitored the pressure with a pressure gauge. The pressure within the tube changed on average by a factor of 1.5 as the tube is quite leaky. Therefore, each measurement is compared to the maximum pressure within its measurement time window. The pressure gauge has an error of 20% due to a lack of recent calibration. The tube does not contain heating or cooling equipment and was left at room temperature (23 °C) limiting the measurements to the Doppler broadened spectra. The absorption coefficient α is calculated by

$$\alpha = -\frac{\ln\left(\frac{I_{H_2O}}{I_0}\right)}{d}, \quad (9)$$

Where I_0 is the intensity of the reference measurement, I_{H_2O} the water vapor measurement and $d = 33$ cm the mean THz path length inside the vacuum tube. " (lines 533-552)

Reviewer #3, Concern #3: The authors should remove the subjective word “excellent” in the added sentence: “By simply sweeping the LO frequency, we can record the spectrum of any unknown source under test at a resolution defined by the excellent spectral purity of the FC-referenced LO.

Author response: We thank the reviewer for spotting subjective wording.

Author action: We rewrote sections with subjective wording to be more objective wherever possible. For a better understanding of the context, we also added a discussion on linewidths of similar THz optical synthesizer (please see reviewer 1, concern 1 for the details).

“The presented setups offer the broadest continuous-wave frequency coverage to date, combined with a sharp spectral resolution, enabling diverse applications ranging from fast non-destructive testing, astronomic high-resolution spectroscopy, to frequency-modulated RADAR.” (lines 24-27)

“Nonetheless, the electronic system benefits from an Hz- to kHz-level spectral resolution and a very low noise floor.” (lines 68-69)

“The laser system has the capability of covering more than 10 THz [32] with competitive phase stability [33] at a Hz-level spectral resolution [34, 35] at the lower end of the THz spectrum while sweeping with maximum speeds of 1 THz/s [32].” (lines 108-110)

“By simply sweeping the LO frequency, we can record the spectrum of any unknown source under test at a resolution defined by the spectral purity of the FC-referenced LO.” (lines 138-140)

“Commercially available electronic spectrum analyzers only reach frequencies up to 1.5 THz. As tab. 1 shows the presented photonic system significantly exceeds the frequency range of the electronic variants while achieving slightly better spectral resolution and comparable noise floors.” (lines 342-345)

“The result is a lower achievable measurement bandwidth in case of a non-referenced CW signal, while the usable measurement bandwidth increases with better relative stability between the SUT and LO enabling a measurement bandwidth of 6.5 THz with all spectral power concentrated within a 2 Hz frequency bin.” (lines 513-516)

Response to reviewers on Nature Communications manuscript NCOMMS-24-72395-B

Dear Reviewers,

We sincerely thank you for thoroughly reviewing the manuscript, the comments and suggestions and for accepting the manuscript. We would still like to take the opportunity to thank each reviewer individually and comment on the final remarks.

Please note that, in preparation for the editorial compliance, we uploaded the figures separately and moved the table and figure descriptions to the end of the manuscript.

Yours sincerely,

Benedikt Krause

(Corresponding author, in the name of all authors)

Reviewer #1: The authors have addressed the reviewer's concern by adding other relevant LO / THz frequency synthesizer methods to the introduction with a short comparison in the revised manuscript. Now this reviewer agrees to accept the manuscript for publication.

Author response: We thank the reviewer for proofreading our manuscript and for the helpful comments in the previous revisions.

Reviewer #2: The authors have thoroughly addressed all reviewers' concerns and well complemented the paper with missing measurements, clarifications, extensions. I am in favor of the publication of the manuscript in Nat Comm.

Author response: We thank the reviewer for proofreading our manuscript and for the helpful comments in the previous revision.

Reviewer #3: I appreciate the effort the authors have made to address the reviewers' comments, and I believe this manuscript merits publication in Nature Communications. That said, I would still encourage the authors to reconsider the claim that the presented THz detection technology can operate effectively beyond 10 THz. As it stands, this assertion is not directly supported by the data in the manuscript, and relies solely on a single conference paper: Proceedings Volume 12885, Terahertz, RF, Millimeter, and Submillimeter-Wave Technology and Applications XVII; 1288503 (2024) While figure 3b from that conference paper does show a THz signal above the noise floor extending to 10 THz, the nature of this signal is ambiguous. Since the spectrum reflects the absolute value of the Fourier-transformed time-domain signal, the high-frequency components, 6 to 7 orders of magnitude weaker than the low-frequency signal, could plausibly arise from minor amplitude fluctuations of a time-resolved signal containing only low frequencies. This contribution is distinct from the "noise" indicated in the figure. In previous experiments, such high-frequency artifacts could be entirely suppressed by applying a spectral filter (high pass) that reduces the low-frequency contribution of the THz signal (e.g. see Fig. 2a in Light Sci Appl 14, 44 (2025)) This is why I find the repeated claim of >10 THz operation mildly problematic. Although not a cause for rejection, reiterating this claim in a Nature Communications article risks prematurely solidifying the notion that this device has been demonstrated to operate in that regime. I am not suggesting the claim is incorrect, only that the supporting evidence is very weak. The concern is that future researchers may cite this work as a definitive proof of >10 THz operation, potentially motivating follow-on studies based on an unverified premise.

Author response: We thank the reviewer for proofreading our manuscript and for the helpful comments throughout all revisions. In fact, this statement as well as the measurement in ref [36] up to 10 THz is not really relevant for the current manuscript and therefore we softened the statement.

Author action: We reworded the claim to not directly mention a specific frequency for the photoconductive mixers:

"Removing the substrate allows for potential coverage within or possibly even beyond the Reststrahlenband of InP and InGaAs [36]." (lines 144-145).

“By removing the substrate the frequency coverage of the photoconductive mixer can be extended, as demonstrated in ref. [36], potentially within or possibly even beyond the Reststrahlenband of InP and InGaAs , to cover a similar range as the FC-referenced photonic LO.” (lines 319-322)

Between the lines:

We agree with the reviewer that stating an unsupported or only mildly supported claim may be mistaken as a definite proof and the authors had seen cases themselves in our own lab as well as false data in publications where an imperfectly corrected time axis led to fake echoes/artifacts at the high frequency end after Fourier transformation. We agree that such artifacts become particularly apparent when there is a high dynamic range at low frequencies as the artifacts are essentially attenuated copies of the low frequency part caused by the FFT that makes the low frequency end reappear at the high frequency end. Yet, we still believe that the statement of potential usability even beyond 10 THz is correct and might be relevant for future work for the following reasons:

- 1.) There is no plausible reason why photoconductors should not work beyond 10 THz.
- 2.) Fourier artifacts can easily be identified by the absence of absorption lines. Fig. 1b of ref [36] shows a TDS measurement of a sample with InP substrate. One can see a perfectly flat noise floor within the Reststrahlenband, starting around 6.5 THz up to ~8 THz where this specific measurement ended. The slight raise of the signal just at 8 THz is indeed supported by the FTIR data. In contrast, when the substrate is removed (Fig 3b), there is no apparent noise at these frequencies, but there still seems to be some mild absorption between 6.5 and 7.8 THz, possibly originating from the photoconductive material’s Reststrahlenband (just an ~1.5 μm thick film, so no excessive attenuation). Indeed, we cannot make a rock-solid statement for frequencies beyond 8 THz as the measurement in Fig. 1b contains no data beyond 8 THz.

Fig.1 b)

Fig. 3b)

Given there is little to no papers with experimental data on photoconductive mixers beyond 10 THz besides from authors within this manuscript, we encourage other groups or authors to investigate the behavior.